# Neural dynamics of the attentional blink revealed by encoding orientation selectivity during rapid visual presentation

Matthew F. Tang [1,2,3]*, Lucy Ford [1], Ehsan Arabzadeh [2,3], James T. Enns [4,5], Troy A.W. Visser[5] & Jason B. Mattingley [1,2,6,7]

The human brain is inherently limited in the information it can make consciously accessible. When people monitor a rapid stream of visual items for two targets, they typically fail to see the second target if it occurs within 200–500 ms of the first, a phenomenon called the attentional blink (AB). The neural basis for the AB is poorly understood, partly because conventional neuroimaging techniques cannot resolve visual events displayed close together in time. Here we introduce an approach that characterises the precise effect of the AB on behaviour and neural activity. We employ multivariate encoding analyses to extract feature-selective information carried by randomly-oriented gratings. We show that feature selectivity is enhanced for correctly reported targets and suppressed when the same items are missed, whereas irrelevant distractor items are unaffected. The findings suggest that the AB involves both short- and long-range neural interactions between visual representations competing for access to consciousness.

[1] Queensland Brain Institute, The University of Queensland, Brisbane, QLD, Australia. [2] Australian Research Council Centre of Excellence for Integrative Brain Function, Victoria, Australia. [3] Eccles Institute of Neuroscience, John Curtin School of Medical Research, The Australian National University, Canberra, ACT, Australia. [4] Department of Psychology, The University of British Columbia, Vancouver, BC, Canada. [5] School of Psychological Sciences, The University of Western Australia, Perth, WA, Australia. [6] School of Psychology, The University of Queensland, Brisbane, QLD, Australia. [7] Canadian Institute for Advanced Research (CIFAR), Toronto, Canada. *email: matthew.tang@anu.edu.au

Despite the remarkable capacity of the human brain, it is found wanting when undertaking multiple tasks concurrently, or when several goal-relevant items must be dealt with in rapid succession. These limitations are particularly evident when individuals are required to execute responses to multiple items under time pressure[1,2], or when they must report relevant target items that appear briefly and in rapid succession[3–5]. Elucidating the source of these limitations has been a persistently difficult challenge in neuroscience and psychology. While the neural bases for these processing limits are not fully understood, it is widely assumed that they are adaptive because they provide a mechanism by which selected sensory events can gain exclusive control over the motor systems responsible for goal-directed action.

Here we address a long-standing question concerning the neural basis of the widely studied attentional blink (AB) phenomenon, where observers often fail to report the second of two target items (referred to as T2) when presented within 200–500 ms of the first target (T1) in a rapid stream of distractors[3–5]. Functional magnetic resonance imaging (fMRI) lacks the temporal resolution to accurately characterise neural activity associated with the rapid serial visual presentation (RSVP) tasks presented at rates of 8–12 Hz, which are commonly used to elicit the AB[6,7]. Even electroencephalography (EEG), which has relatively good temporal resolution, produces smeared responses to items in an RSVP stream[8]. Furthermore, mass-univariate approaches applied to fMRI or EEG data only measure overall neural activity while providing no information about how neural activity represents featural information carried by single items (e.g., their orientation).

Here we overcome these limitations by combining recently developed multivariate modelling techniques for neuroimaging[9–16] with an RSVP task designed to determine the neural and behavioural basis for the AB. Forward (or inverted) encoding modelling determines the neural representation of feature-selective information contained within patterns of brain activity, using multivariate linear regression. This approach allowed us to explicitly measure the neural representation of specific features—in this case, orientation-selective information elicited by grating stimuli—separately for each item within an entire RSVP stream.

We use this approach to address two central theoretical questions. First, does selection of a target from within an RSVP stream increase the gain or the precision of its neural representation? Previous efforts to answer this question in the domain of spatial attention have come from single cell recordings in non-human primates[17,18], as well as whole-brain activity measured using fMRI and EEG in humans[15,19]. With few exceptions, these studies have found that spatial attention increases the gain of feature-selective processing of attended items. By contrast, feature-based manipulations of attention, in which specific characteristics of an item such as its colour or motion are cued for selective report, typically result in a sharpening of neural selectivity[20,21]. To date, it remains unknown whether the limits of temporal attention in the AB are associated with changes in neural tuning to targets, distractors, or both classes of items. The neural response in human primary visual cortex[6] and macaque lateral intraparietal area[22] to the second target is reduced overall on AB trials compared with non-AB trials, while subtraction-based EEG designs have shown that a late-stage component of the ERP (the N400) is reduced 200–400 ms after target presentation[8]. Critically, however, these measures cannot determine how the AB affects the neural representation of visual information, which could conceivably reflect a reduction in gain, an increase in tuning sharpness, or both.

A second, unresolved theoretical question concerns the source of the AB. Existing theories have often attributed the AB to either extended processing of the first target, or to inadvertent distractor processing. In the first class of theories, it is assumed that all items generate representations in early visual areas, but that the system inhibits items after T1 detection to avoid contamination by distractors[23–26]. On other accounts (so-called 'distractor-based' theories), the AB is assumed to reflect a cost associated with switching between target and distractor processing[27]. Finally, a third class of theories argues that the representation of the second target can become merged with either the first target or the distractors[28,29]. This class of theories is motivated by the finding that the perceived order of targets is often reversed (i.e., T2 is reported as appearing before T1).

Our RSVP task consists of a stream of randomly oriented Gabor gratings, with two higher spatial-frequency targets set amongst lower-spatial-frequency distractors (Fig. 1a and Supplementary Movie 1). At the end of the stream, participants are

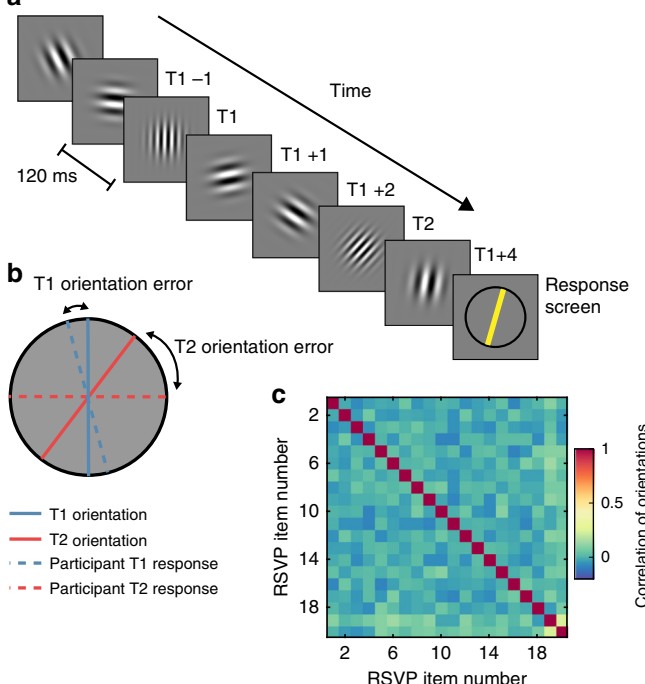

**Fig. 1 Schematic of stimuli and timing of displays in the rapid serial visual presentation (RSVP) task. a** An illustration of a typical trial in the RSVP task, which consisted of 20 sequentially presented Gabor patches at fixation. Each of the twenty items within a single RSVP stream was presented for 40 ms, with an 80 ms blank interval between items (120 ms inter-stimulus interval), yielding an 8.33 Hz presentation rate. The number of items (Lag) between the first (T1) and second (T2) targets was varied to measure the temporal duration of the AB. At the end of each RSVP stream, participants reproduced the orientations of T1 and T2 (higher spatial-frequency gratings) in the order in which they were presented by adjusting an on-screen cursor at the end of the trial. They were asked to determine the orientations as accurately as possible and were not given any time restriction to do this. Visual feedback was provided following the response. **b** A schematic of the feedback screen for responses. **c** The correlation values between orientations of the RSVP items over trials in Experiment 1. As Gabor orientations were randomly drawn (without replacement) on each trial, across all trials the orientation of any given item in the stream was uncorrelated with the orientation of any other item. This permitted the use of regression-based approaches to isolate the behavioural and neural processing of individual items independently of surrounding items within the stream. The correlations were calculated for each participant and are displayed as averaged across participants.

asked to reproduce the orientations of the two targets (Fig. 1b). Critically, the orientation of each item in the stream is uncorrelated with the orientation of all other items (Fig. 1c), thus permitting the use of linear regression analyses to separately extract the influence of each item in the stream on neural activity measured by EEG, and on behavioural reports of the orientations of the two targets. These aspects of the experimental design allowed us to quantify the influence of both targets and distractors on participants' perceptual reports and on their associated neural representations.

To preview the results, the behavioural target task replicated the hallmarks of the AB effect: the orientation of T1 was reported with a relatively high degree of accuracy, whereas orientation judgements for T2 were degraded when T2 appeared 200–400 ms after T1. Forward encoding analyses of EEG activity showed that targets evoked greater orientation-selective information than distractors when T2 was accurately reported (i.e., in non-AB trials), and that orientation information evoked by T2 was suppressed, relative to the distractors, when T2 was missed (i.e., in AB trials). Critical to our first question of whether focused attention influences the gain or precision of feature-specific

representations, only the gain of the encoded EEG response was affected by T2 response accuracy.

With respect to our second question—whether accuracy in registering the second target is linked to the processing of T1 or to the intervening distractors—the evidence was in favour of T1-based theories of the AB. We found no evidence to suggest that neural representations of the distractors are affected by the AB. Finally, we describe an unexpected observation—one not predicted by any theory of the AB—namely, a significant interaction between the specific features of T1 and T2, implying a previously unknown long-range temporal integration of target representations within rapid sequential visual streams.

## Results

**Experiment 1—behavioural hallmarks of the AB.** Participants' ($N = 22$) response errors (i.e., the difference between the presented and reported orientation for each target) were centred around 0°, verifying that they were able to perform the task as instructed. Figure 2a captures the temporal dynamics of the AB, such that accuracy was affected by target position (T1 or T2) and

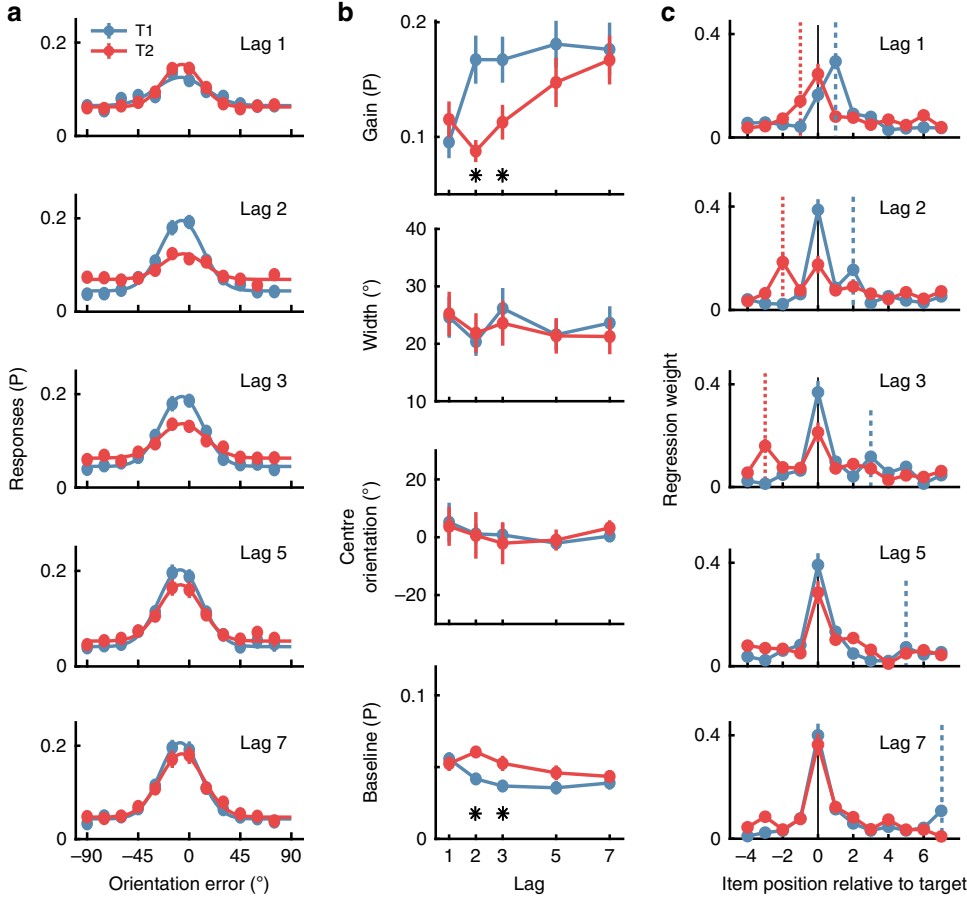

**Fig. 2 Behavioural results for the RSVP task in Experiment 1. a** The distribution of response errors (difference between presented and reported orientation) across participants ($N = 22$) for the first (T1, blue lines) and second (T2, red lines) target for each Lag condition. The line shows fitted four-parameter Gaussian function. **b** Quantified behavioural responses for the four parameters of the fitted Gaussian function (see Supplementary Fig. 1) for each participant. Gain shows the amplitude, width shows the standard deviation of the function, centre orientation is the mean (which should be centred around 0° for unbiased estimates), and baseline is a constant parameter accounting for non-orientation selective responses which indicates guessing. Asterisks indicate Bonferroni-corrected *t*-tests showing significant differences at $p < 0.05$. **c** Regression results for the influence of distractors and targets on participants' responses. Higher regression weights indicate that a given item's orientation was more influential for determining the reported orientation. The dotted vertical lines indicate the position of the other target (colour matched). Consider, for example, the panel depicting Lag 2 results. For T1 report, T2 occurred at item plus 2 as indicated by the dotted blue line, whereas for T2 report, T1 occurred at item minus 2, as indicated by the dotted red line. Across all panels, error bars indicate ∓1 standard error of mean.

Lag. Specifically, at Lag 1 accuracy for both T1 and T2 was degraded relative to accuracy at the other lags (2, 3, 5 and 7). Moreover, at Lags 2 and 3, T1 accuracy was high whereas T2 accuracy was relatively poor. This was largely due to an increase in the baseline guessing rates (where errors occurred evenly across all orientations). Finally, at longer temporal separations (Lags 5 and 7), target accuracy was similar for both items.

**Experiment 1—modelling the AB using behavioural data**. We fitted Gaussian functions to each individual's data to quantify how the AB affected target perception (Fig. 2b; see Methods and Supplementary Fig. 1). The accuracy reduction for T2 at Lags 2 and 3 was primarily linked to a reduction in gain. A 2 (Target; T1, T2) × 5 (Lag; 1,2,3,5,7) within-subjects ANOVA showed the gain parameter was affected by Target ($F(1,21) = 10.00$, $p = 0.005$, $\eta_p^2 = 0.32$) and Lag ($F(4,84) = 11.66$, $p < 0.0001$, $\eta_p^2 = 0.36$), and the interaction between these factors ($F(4,84) = 7.10$, $p < 0.0001$, $\eta_p^2 = 0.25$). Critically for our first theoretical question, the spread (width) of orientation errors was unaffected by the factors of Target ($F(1,21) = 0.10$, $p = 0.76$, $\eta_p^2 = 0.005$) or Lag ($F(4,84) = 0.55$, $p = 0.70$, $\eta_p^2 = 0.03$), or by the interaction between these factors ($F(4,84) = 0.19$, $p = 0.94$, $\eta_p^2 = 0.01$). The baseline parameter, which reflects guessing of random orientations, was also significantly affected by the factors of Target ($F(1,21) = 12.72$, $p = 0.002$, $\eta_p^2 = 0.38$) and Lag ($F(4,84) = 4.82$, $p = 0.002$, $\eta_p^2 = 0.19$), and by the interaction between them ($F(4,84) = 5.04$, $p = 0.001$, $\eta_p^2 = 0.19$). These same effects were also evident when the data were not normalised (Supplementary Fig. 2), and with a wide range of parameters to specify the orientation errors (Supplementary Fig. 3).

Taken together, these results are consistent with a previous AB study using similar analysis methods[30]. They also lend weight to the global workspace theory of consciousness in the AB[31], which argues that participants either see the target and have full awareness of it (allowing them to respond precisely), or they have no awareness (and so simply guess randomly). By contrast, the results are inconsistent with the opposing view that the AB involves a noisier (i.e., weaker precision) signal for the target that is inaccurately reported[32].

**Experiment 1—targets, not distractors, influence orientation judgements**. To evaluate the influence of distractors on participants' reports, we aligned the orientations of the items relative to target position within the RSVP stream (−4 to +7 items) and constructed a regression matrix to predict the behavioural response for each target. If the orientation of an item is influential in determining the reported orientation, the regression weight will be relatively high (Fig. 2c). As expected, for all lags, each reported target orientation was influenced principally by its own orientation. The one exception was the item at Lag 1, where the reported orientation of T1 was as strongly influenced by the orientation of T2 as by the orientation of T1. This observation is in line with numerous studies which have suggested that temporal order information can be lost for consecutive targets[29,33]. This phenomenon, also known as Lag 1 switching, where the perceived order of the targets is reversed, explains why the accuracy of orientation judgements on both T1 and T2 was reduced at Lag 1 (see also Supplementary Fig. 4). By contrast, for items at Lags 2 and 3, orientation judgements on T1 were only marginally influenced by the orientation of T2 (i.e., for items at positions +2 and +3, respectively, in the RSVP stream). However, at these same lags (where the AB was maximal) T2 reports were significantly influenced by T1 orientation (i.e., for items at positions −2 and −3, respectively). Importantly, there was no reliable influence of distractors on reported target orientation at any lag,

suggesting distractors played little or no role in target orientation errors.

**Experiment 1—long-range integration of target orientations**. One account of the AB[28,29] has suggested that successive targets presented at short lags are integrated into a single episodic trace, which accounts for Lag 1 switching. With the present task, we can directly quantify how targets are integrated by looking for systematic biases in the reported orientation of a given target based on its orientation difference with respect to the other target. Figure 3a shows orientation judgement errors as a function of the difference between the two target orientations. While the average orientation error is centred on 0°, the perceived orientation of either target (T1 or T2) was significantly biased toward the orientation of the other target within the RSVP stream at early Lags. Furthermore, these biases were orientation-tuned, such that the largest bias occurred when targets differed by approximately 45°, somewhat analogous to serial dependency effects[34,35]. This profile of biases suggests response integration, rather than replacement, as the latter would predict that only the orientation of T2 should drive the reported orientation of T1. Instead, and consistent with our linear regression analysis (see Fig. 2c), the bias reflected the difference between target orientations, which supports the idea that the critical features of the two targets are assimilated over time[28,29].

We fit first derivative of Gaussian (D1) functions[36–38] to quantify the amount of orientation-selective bias for both targets at each Lag for each participant. A 2 (Target; T1, T2) × 5 (Lag; 1,2,3,5,7) within-subjects ANOVA revealed significant main effects of Target ($F(1,21) = 5.04$, $p = 0.04$, $\eta_p^2 = 0.19$) and Lag ($F(4,84) = 6.54$, $p < 0.0001$, $\eta_p^2 = 0.24$), and a significant interaction ($F(4,84) = 6.14$, $p < 0.0001$, $\eta_p^2 = 0.27$). For T1 reporting, the bias was significantly greater than chance at all intervals, whereas for T2, there was a significant bias at Lags 2 and 3 only (Bonferroni-corrected one-sample $t$-test, all $ps < 0.05$). As might be expected[28,29], the 'attraction' bias in target reports was strongest when the two targets were presented with no intervening distractors between them (i.e., at Lag 1). An entirely unexpected finding, however, is that there was an equally strong attraction bias between targets presented at Lags 2 and 3 (see Fig. 3b), even though participants were not explicitly aware of the orientation of T2 on AB trials.

**Experiment 1—biased perception of targets by preceding distractors**. Previous work suggests that distractor processing can significantly interfere with target processing[39–41], particularly for the immediate post-target item which can be integrated into the target representation[28,29,33]. To determine whether this was the case in our data, we repeated the previous analysis but used the difference in orientation between the target and each of the other items in the RSVP stream (Fig. 3c). For most lags, the reported target orientation was significantly attracted toward the immediately following distractor, but was not reliably influenced by any other distractor. A 2 (Target; T1, T2) × 5 (Lag; 1,2,3,5,7) × 5 (Item position; −1,1,2,3,4) within-subjects ANOVA confirmed a significant three-way interaction between the factors ($F(16,336) = 4.11$, $p < 0.0001$, $\eta_p^2 = 0.16$). At Lag 1, there was no influence of distractors on reported orientations for either T1 or T2. Taken with the previous result, this suggests that the representation of a given target is influenced by both the other target and by the post-target item. The results suggest that when the visual system detects a target, it automatically integrates features from the immediately subsequent item. This is consistent with previous studies that have highlighted the importance of masking by the item immediately following the target in eliciting the AB[42].

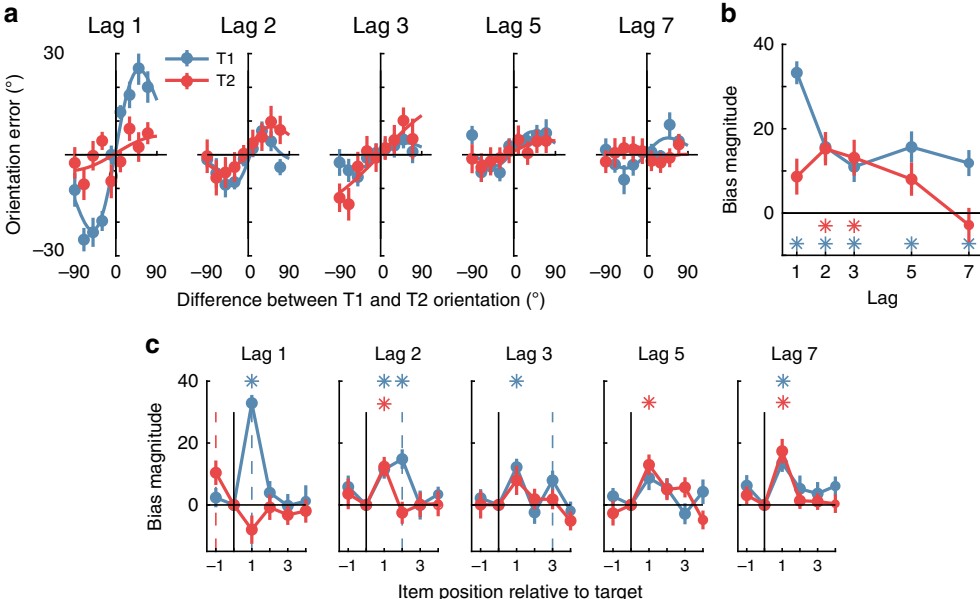

**Fig. 3 The influence of targets and distractors on the reported orientation in Experiment 1. a** Orientation error (the difference between presented and reported orientation) plotted against the difference between T1 (blue lines) and T2 (red lines) orientations (divided into 30° bins, for clarity of presentation). Positive values on the X-axis indicate that a given target was rotated clockwise relative to the other target. For instance, when examining T1, a positive value indicates that T2 was rotated clockwise relative to T1, whereas a negative value indicates that T2 was rotated anti-clockwise relative to T1. For T1, the plotted values reflect the calculation of T1 minus T2, and vice versa for the calculation of T2, to ensure values were equivalent for the comparison of interest. The same convention applies to orientation error, shown on the Y-axis. The fitted line is the first derivative of a Gaussian (D1) function showing the orientation-tuned gain and width of the response. **b** Bias magnitude was quantified across participants by fitting the D1 function to each participant's (non-binned) data, with the gain showing bias magnitude. A positive gain on the Y-axis indicates that the perceived orientation was biased toward, rather than away from, the other target. **c** Bias magnitude by difference with target and distractors. For both T1 and T2, the difference between the target and the item was found in the same manner as in (**a**). We fit the D1 function to find the magnitude of bias induced by each item for each participant. The dotted coloured lines indicate the temporal position of the other target (T1 = blue, T2 = red). For all panels, the asterisks indicate, for each target (colour matched), at which lags the bias was significantly greater than zero (Bonferroni-corrected one-sample t-tests p < 0.05). Across all panels, error bars indicate ∓1 standard error of mean.

**Experiment 2—electrophysiological recording of the AB.** We next characterised the neural activity elicited by individual RSVP items, and determined how this was affected by the AB. In Experiment 2, a group of 23 new participants undertook the RSVP task introduced in Experiment 1 while neural activity was concurrently measured using EEG. The method was identical in all respects, except that we now included targets only at Lags 3 and 7 (i.e., a single target inside and outside the AB, respectively) to increase the within-subject power for the EEG analyses.

**Experiment 2—behavioural results.** The behavioural results replicated, in all important respects, those found in Experiment 1. As shown in Fig. 4a, participants performed well overall, and their orientation judgements for T1 and T2 were centred on the presented orientations. As in Experiment 1, we fit Gaussian functions to quantify the results (Fig. 4b). For the gain parameter, a 2 (Target; T1, T2) × 2 (Lag; 3, 7) within-subjects ANOVA revealed significant main effects of Target ($F(1,22)$ = 11.63, $p = 0.003$, $\eta_p^2 = 0.35$) and Lag ($F(1,22) = 18.70$, $p < 0.0001$, $\eta_p^2 = 0.46$), and a significant interaction ($F(1,22) = 40.19$, $p < 0.0001$, $\eta_p^2 = 0.65$). Likewise for the baseline parameter, there were significant effects of Target ($F(1,22) = 8.96$, $p = 0.007$, $\eta_p^2 = 0.30$) and Lag ($F(1,22) = 12.21$, $p = 0.002$, $\eta_p^2 = 0.36$), and a significant interaction ($F(1,22) = 7.91$, $p = 0.01$, $\eta_p^2 = 0.26$). By contrast, there were no significant main effects and no interaction for the width parameter (Target ($F(1,22) = 1.19$, $p = 0.29$, $\eta_p^2 = 0.05$; Lag ($F(1,22) = 3.90$, $p = 0.06$, $\eta_p^2 = 0.15$); interaction ($F(1,22) = 0.14$, $p = 0.71$, $\eta_p^2 = 0.006$).

**Experiment 2—orientation selectivity of RSVP items.** We next applied forward modelling to the EEG data recorded during the task to quantify orientation information contained within multivariate patterns of neural activity. Because the orientations of successive items were uncorrelated, we were able to quantify orientation selectivity for each grating without contamination from adjacent items. Forward encoding uses a linear regression-based approach to find multivariate patterns of EEG activity that are selective for features of interest—in this case orientation. As no previous study has used forward encoding in conjunction with rapid visual presentations, we first verified that orientation selectivity for each of the 20 RSVP items could be extracted separately using this approach, and at what time point any such response was evident. To do this, we constructed 20 encoding models, one for each of the item positions within the 20-item RSVP stream, based on the orientations presented for that item across trials.

As shown in Fig. 5, the forward encoding revealed robust and reliable feature selectivity derived from patterns of EEG activity for each of the gratings presented during the RSVP. Each item's orientation was successfully decoded over a time window that extended from 74 to 398 ms after the item was presented. Examination of the neural responses to each of the 20 items within the RSVP stream (Fig. 5c) shows that feature selectivity was evident as a series of regularly spaced, short-lived impulse responses, each with a delay of around 50 ms from grating onset and lasting approximately 300 ms. To quantify these observations, we fit Gaussian functions to the forward encoding results for each item separately for each participant and at each time

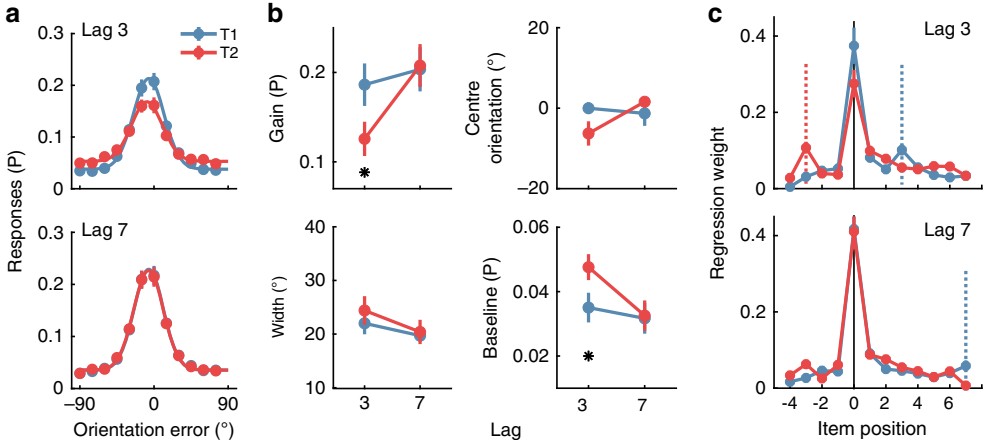

**Fig. 4 Behavioural results for the RSVP task in Experiment 2. a** Aggregate response accuracy across participants (difference between presented and reported orientations) for T1 (blue lines) and T2 (red lines), shown separately for Lag 3 and Lag 7 trials. Lines are fitted Gaussian functions. **b** Quantified behavioural responses for the four parameters of the fitted Gaussian functions (gain, width, centre orientation and baseline) to each participant's data. Asterisks indicate Bonferroni-corrected t-test differences at $p < 0.05$. **c** Regression results for the influence of distractors and targets on participants' responses. The dotted vertical lines indicate the position of the other target (colour matched). Consider, for example, the panel depicting Lag 3 results. For T1 report, T2 occurred at item plus 3 as indicated by the dotted blue line, whereas for T2 report, T1 occurred at item minus 3, as indicated by the dotted red line. Across all panels error bars indicate ∓1 standard error of mean.

point. There was significant feature selectivity (given by the gain of the Gaussian) for each item immediately after it was presented but not before (Fig. 5d). These representations were temporally overlapping, such that multiple orientation-selective responses (~3) were detectable at the same time. Taken together, the forward encoding analyses verify that it is possible to reliably recover the presented orientation of every RSVP item from the multivariate pattern of neural activity recorded using EEG.

### Experiment 2—reduced feature-selective information for T2 during the AB.

We next examined how neural representations of the target items were affected by the AB. To increase signal-to-noise for training the encoding model, we aligned the EEG data to the presentation time of each item in the RSVP stream and applied the same forward encoding procedure. This meant that the model was trained and tested across 12,000 presentations (600 trials by 20 RSVP items; see Fig. 6). To determine the effect of the AB on orientation-selectivity, we separated the forward encoding results by target (T1,T2) and T2 accuracy (correct, incorrect). For the purposes of the analyses, trials were scored as correct if the reported orientation was within ±30 degrees of the presented orientation, a criterion which yielded roughly equal correct and incorrect trials at Lag 3. In line with the AB literature, for all the EEG analyses we only included trials where participants correctly identified T1. Applying these criteria yielded the classic AB effect (Supplementary Fig. 5). A 2 (Lag; 3, 7) × 2 (Target; T1, T2) within-subjects ANOVA applied to these scores revealed significant main effects of Lag ($F(1,22) = 19.05$, $p < 0.0001$, $\eta_p^2 = 0.46$) and Target ($F(1,22) = 18.00$, $p < 0.0001$, $\eta_p^2 = 0.45$), and a significant interaction ($F(1,22) = 31.91$, $p < 0.0001$, $\eta_p^2 = 0.59$). Follow-up t-tests showed that Lag 3 accuracy was significantly lower than Lag 7 accuracy for T2 items ($t(22) = 5.20$, Bonferroni $p = 0.0001$, $d = 0.44$) but not for T1 items ($t(22) = 2.11$, Bonferroni $p = 0.09$, $d = 0.44$). In addition, T2 accuracy was significantly lower than T1 accuracy at Lag 3 ($t(22) = 5.94$, Bonferroni $p < 0.0001$, $d = 1.08$), but there was no such difference at Lag 7 ($t(22) = 1.20$, Bonferroni $p = 0.48$, $d = 0.25$).

We again fitted Gaussians to each time point to quantify the amount of feature-selective information evoked by the targets. For both T1 and T2, there was significant feature-selective activity

shortly after each item appeared (Fig. 6a). For Lags 3 and 7, there was no difference between correct and incorrect trials for the T1 representation. For T2, however, incorrect trials resulted in a significantly decreased feature-selective response (cluster $p = 0.02$) relative to correct trials shortly after each item appeared (100–150 ms) at Lag 3, although the response was not completely suppressed. There were no significant differences in the orientation-selective response between correct and incorrect trials for T2 at Lag 7, suggesting the suppression is caused by the AB rather than general target detection. This was expected because the AB typically lasts less than 500 ms, and is consistent with the current behavioural results showing an AB at Lag 3 but not at Lag 7. Performing the same analysis on the other parameters of the Gaussian (width, centre, baseline) showed no effect of the AB (Supplementary Fig. 6).

To ensure we did not miss any small but consistent effects, we averaged the forward encoding results (Orientation × Time) over the early (100–150 ms) timepoints to increase signal-to noise-ratio and recovered the orientation tuning curve (Fig. 6b). Fitting Gaussians to these values confirmed that the AB was associated with a change in the gain of feature selectivity for T2 at Lag 3, such that correct trials showed significantly greater gain than incorrect trials ($t(22) = 3.12$, $p = 0.01$, $d = 0.65$; Fig. 6b upper panel). By contrast, the width of the representation was again unaffected by the AB ($t(22) = 1.66$, $p = 0.11$, $d = 0.35$) for the same item. For Lag 7 items, neither the gain ($t(22) = 0.12$, $p = 0.90$, $d = 0.03$; Fig. 6b lower panel) nor the width ($t(22) = 0.04$, $p = 0.96$, $d = 0.01$) of the neural representations of T2 items were affected by behavioural performance (correct vs. incorrect trials).

The reduction in T2 selectivity for incorrect trials at Lag 3 was not driven by an arbitrary split of trials into correct and incorrect categories. To verify this, we sorted the evoked T2 forward encoding results by the amount of orientation error (in 15° error bins to allow sufficient signal-to-noise ratios for fitting). There was significantly greater feature selectivity when the orientation error was small, and this selectivity gradually decreased with larger errors (one-way within-subjects ANOVA, $F(1,22) = 2.76$, $p = 0.02$, $\eta_p^2 = 0.11$; Fig. 7). Note that this finding is inconsistent with a graded model of the AB, and instead supports the idea that response variability during the AB is associated with both a decrease in feature-selective gain and an increase in the rate of

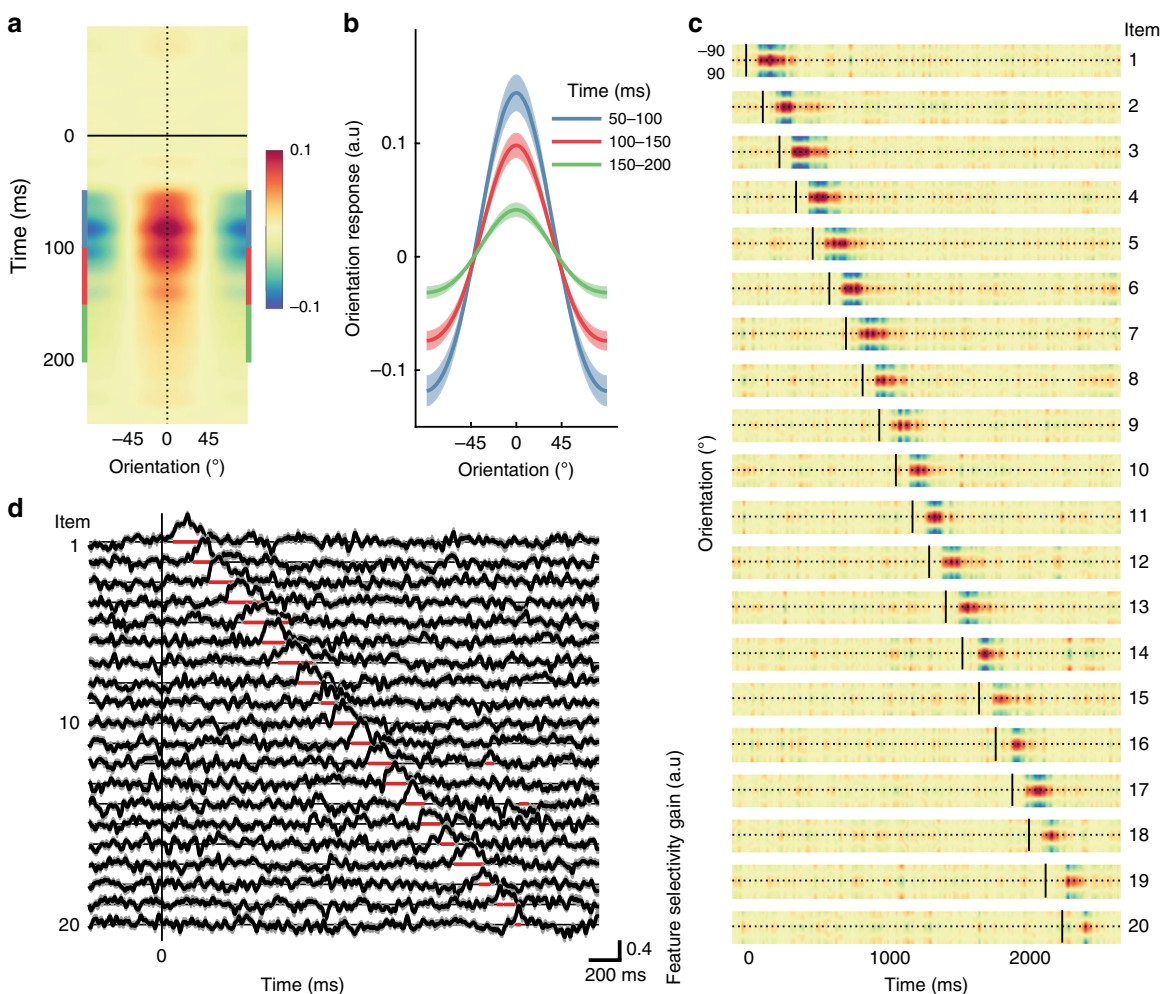

**Fig. 5 Feature (orientation) selectivity for RSVP items derived using forward encoding modelling of multivariate patterns of EEG activity in Experiment 2. a** Forward encoding results aligned at the time of item onset and the presented orientation across all participants. All representations have been re-aligned so that the presented orientation is equivalent to 0°. **b** Forward encoding results averaged over 50 ms bins (shown by corresponding colour in (**a**) following each item. Feature-selectivity peaks around 50–120 ms after the onset of each item and persists for ~200 ms. **c** Forward encoding results for each item in the RSVP stream. Vertical black lines indicate the presentation time of each of the 20 items within the RSVP stream. The dotted horizontal line indicates the presented orientation. The colour scale is the same as in panel (**a**). **d** Gaussian distributions were fitted to each participant's data for each item in the stream, with the gain showing feature selectivity. The red horizontal line segments underneath each trace indicate timepoints over which feature selectivity was significantly different from zero (i.e., where feature selectivity was greater than what would be expected by chance; two-tailed, sign-flipping cluster-permutation, alpha $p < 0.05$, cluster alpha $p < 0.05$, $N$ permutations = 20,000), which occurs immediately following item presentation. Across all panels shading indicates ∓1 standard error of the mean across participants. a.u. = arbitrary units.

guessing. This finding is consistent with the behavioural results, which suggest a discrete model of the AB. Overall, these results indicate that the AB is associated with a reduction in gain, but not width, of feature-selective information for the second target item (T2), and that this effect occurs soon after the target appears within the RSVP stream.

**Experiment 2—only targets affect the AB, but not distractors.** We next examined the neural representations both of targets and distractors to test the different predictions made by T1-based[25,26] versus distractor-based[27] accounts of the AB. T1-based accounts argue that the second target deficit is caused by extended processing of the first target, whereas distractor-based accounts, argue that deleterious processing of the distractors, mainly between T1 and T2, causes the second target to be missed. The theories thus make distinct predictions about the neural representation of target and distractor items. According to T1-based

accounts, target representations should be enhanced relative to those of distractors, and missed T2 items on AB trials should be more poorly represented than correctly reported T2 items. By contrast, distractor-based accounts predict that neural representations of distractor items should be stronger on AB trials than on non-AB trials and weaker following T1 presentation.

As before, we averaged the forward encoding modelling representations (Orientation × Time) across an early time point (100 to 150 ms), and fit Gaussians to each participant's data to quantify feature selectivity (Fig. 8a). For correct trials (i.e., orientation responses to T2 were within 30° of the presented orientation), the two targets resulted in significantly higher feature selectivity (gain) than the immediately adjacent distractors ($-2, -1, +1$ and $+2$ items) for both T1 and T2 representations (all $ps < 0.04$). On incorrect trials, feature selectivity for T1 was not significantly greater than selectivity for the surrounding distractors ($t(22) = 0.15$, $p = 0.88$, $d = 0.03$), even though we included only trials in which T1 was correctly reported. Most

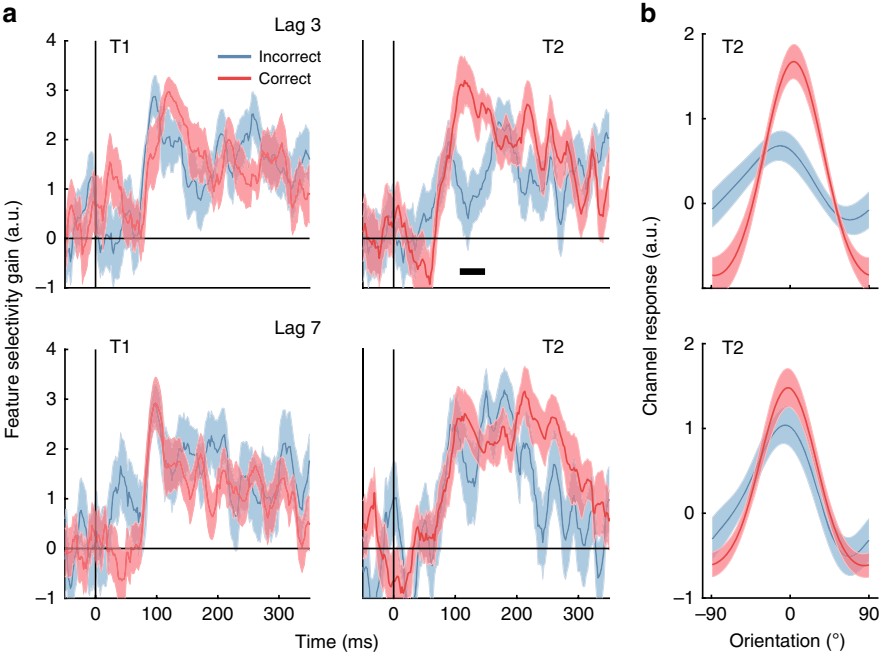

**Fig. 6 Neural representations of feature-selective information during the AB for the first (T1) and second target (T2) based on reporting accuracy for T2 for Experiment 2. a** Time course of measured feature selectivity for T1 and T2, given by the gain of the fitted Gaussian parameter. Trials were scored as correct if the participant's response was within 30° of the presented orientation. Only trials in which participants responded accurately to T1 were included in the analysis. The thick black horizontal line in the upper right panel indicates a period of significant difference between Incorrect (blue lines) and Correct (red lines) trials (two-tailed sign-flipping cluster-permutation, alpha $p < 0.05$, cluster alpha $p < 0.05$, N permutations = 20,000). Note that the difference in magnitude for the encoding results shown in Fig. 5 is due to the increased number of training trials used in this analysis (12,000 vs 600). **b** Forward encoding results were averaged across the significant timepoints for T2 Lag 3 shown in (**a**) (upper right panel) to reconstruct the full representation of orientation. Reliable changes in the gain of orientation representations for T2 were present at Lag 3 (upper panel) but not at Lag 7 (lower panel). There was no difference in the width for either Lag. Shading indicates ∓1 standard error of the mean.

interestingly, on incorrect trials the representations of T2 items were significantly lower than those of the immediately adjacent distractors ($t(22) = 2.09$, $p = 0.04$, $d = 0.44$), suggesting that the featural information carried by T2 was suppressed, while distractors were unaffected. To directly test the distractor model of the AB, we compared distractor representations before T1 with distractor representations during the AB (i.e., between T1 and T2). The account predicts that distractors presented during the AB should elicit a stronger neural representation as they are likely to be incorrectly selected as targets. Instead, we found that distractors were represented similarly before and during the AB for both correct trials ($t(22) = 0.85$, $p = 0.40$, $d = 0.18$) and incorrect trials ($t(22) = 1.83$, $p = 0.08$, $d = 0.38$). Taken together, these results suggest that for trials where participants accurately report target orientation, the neural representations of targets are boosted relative to those of distractors. By contrast, when the second target is missed, as occurs during the AB, there is a significant suppression of the target's featural information.

**Experiment 2—localisation of feature selectivity for targets and distractors**. In a final step, we performed a univariate sensor-level analysis for feature selectivity[10] to find the topographies associated with target and distractor processing. To do this, we trained a simplified model of feature selectivity on each type of item (targets and distractors) separately for each EEG sensor. Orientation information for both targets and distractors was evident most strongly over occipital and parietal areas, and target items generated significantly greater selectivity over these areas than distractors (Fig. 8b). These findings suggest that while target and distractor items are processed in overlapping brain regions,

targets generate significantly greater orientation-selective information than distractors.

## Discussion

We developed an RSVP paradigm to determine the neural and behavioural bases of the limits of temporal attention. The behavioural results replicated the hallmark of the AB with response accuracy being significantly reduced when T2 was presented within 200–400 ms of T1. We discovered that target representations influenced one another, such that the reported orientation of one target was biased toward the orientation of the other. Results from Experiment 2 revealed that successfully reporting T2 depended on a boost to its neural representation relative to other items in the RSVP stream, whereas missing T2 corresponded to a suppressed neural response relative to the distractors. Notably, there was no evidence for suppression of neural representations of the distractors, suggesting the AB is primarily driven by processing competition between target items. This observation supports theories that have attributed the second-target deficit to first target processing[4,23,43], but is inconsistent with theories that attribute the AB to inadvertent processing of distractor items[24,27].

An important but unexpected result is that target reports were influenced by one another despite being separated by several hundred milliseconds and multiple distractor items. One influential theory argues that the AB is caused by temporal integration of the target with the immediate post-target distractor[28,29]. Our RSVP task found evidence for this but also showed that target representations appear to be integrated with each other even when they are separated by multiple distractor items within the stream. This finding is not explicitly predicted by any existing

account of the AB. The largest bias was for Lag 1 trials, in which the two targets appear sequentially, a result that is consistent with Lag 1 switching[28,29,33]. The orientation of the immediate post-target distractor also significantly biased the perceived target orientation, whereas the distractors that appeared between the

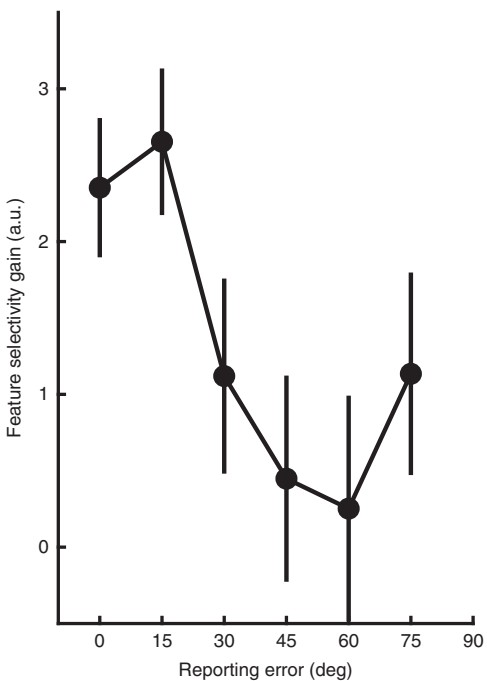

**Fig. 7 Gain of feature-selective information for T2 items presented at Lag 3 in Experiment 2, plotted as a function of reporting error.** Forward encoding results were averaged across early timepoints (100–150 ms), and were binned by the absolute difference between the presented and reported orientations (in 15° increments). Each bin is displayed as the starting value (e.g., 0° incorporates errors from 0° to 15°). Gaussians were fitted to quantify selectivity with the gain parameter shown here. Feature selectivity was highest when participants reported the orientation to within 30° of the presented orientation, and declined significantly with larger reporting errors. Error bars indicate ∓1 standard error of the mean.

targets did not bias perceptual judgements. Taken together, our findings across two experiments suggest that the detection of a target in an RSVP sequence starts a period of local integration which involuntarily captures the next item, whether it is a target or a distractor. This is followed by a more global integration of targets, possibly within working memory[4].

Our first major aim was to determine how the AB affects target representations. The forward encoding modelling of the EEG data adds to previous results[30] by demonstrating that the gain in neural representations of Lag 3 items is significantly reduced in AB trials, compared with non-AB trials. Supporting the behavioural results, there was no effect on the width of EEG-derived feature selectivity during the AB. The neural results also go beyond the behavioural findings by showing that the gain of Lag 3 items is not only suppressed on AB trials, but boosted on non-AB trials compared with those of the distractors. Taken together, these results suggest that temporal attention operates in a similar manner to spatial attention[15,17–19], but not to feature-based attention[20,21], as the former has been found to affect the gain of neural responses whereas the latter tends to affect the sharpness of neural tuning.

The second major aim of our study was to resolve the persistent debate between T1- and distractor-based theories of the AB[4,23–27,43,44]. Behaviourally, we found scant evidence that distractors (apart from the immediately subsequent distractor) influence target perception. Consistent with T1-based accounts of the AB[4,25], there were robust neural representations of distractors and no evidence that distractor representations were boosted following initial target detection, as would be predicted by distractor-based accounts. Furthermore, we found no evidence that post-T1 distractors were suppressed, as would be predicted by T1-based inhibition accounts of the AB[4,23]. Instead, consistent with T1-based accounts, the representations of both targets were boosted relative to those of the distractors. If the second target was missed, however—as occurs during the AB—then the representation of the second target was significantly suppressed relative to the distractors. Taken together, these results suggest that when the first target is processed rapidly, attention is efficiently redeployed to the second target, causing its representation to be boosted. By contrast, if the second target appears while processing of the first target is ongoing, the visual system actively

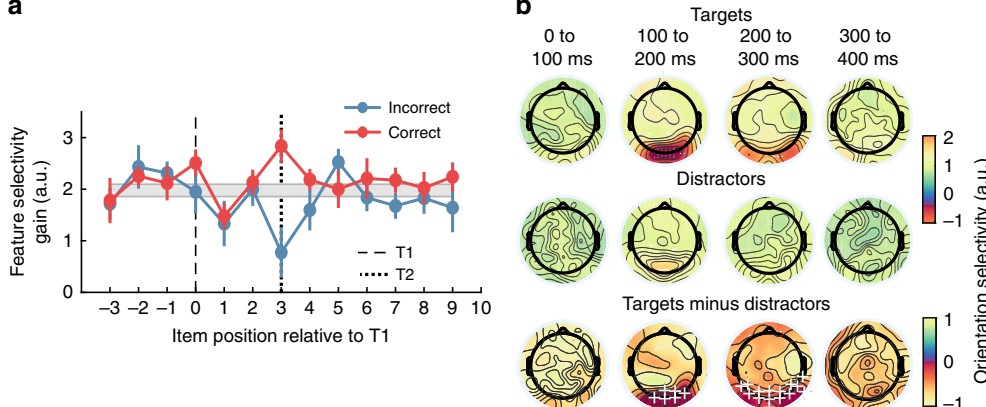

**Fig. 8 Feature selectivity and scalp topographies for targets and distractors in Experiment 2. a** Neural feature selectivity (gain of Gaussian) of target and distractor representations for Lag 3. Blue lines show incorrect trials and red lines show correct trials. Gaussians were fit to the averaged neural representation from 100 to 150 ms. To aid comparison, the grey bar indicates the average distractor representation (∓1 standard error of mean). Note that all distractors and targets have gain values significantly above 0 arbitrary units (a.u.) indicating robust feature selectivity. Error bars indicate ∓1 standard error of mean. **b** Headmaps showing univariate orientation selectivity over time, plotted separately for targets and distractors. Plus symbols indicate positive cluster-permuted differences between targets and distractors (two-tailed cluster-permutation, alpha $p < 0.05$, cluster alpha $p < 0.025$, N permutations = 1500).

suppresses the information to avoid the targets interfering with each other.

Suppression of the T2 representation occurred 100–150 ms after the target appeared, suggesting inhibition of the sensory information by ongoing processing of T1. This fits well with previous work showing that the AB is associated with a reduced late-stage response, as indicated by an ERP component associated with working memory consolidation[8,45]. Taken together with the current results, it appears that the AB is associated with an early suppression of sensory information associated with the T2 stimulus. The diminished strength of sensory information associated with the T2 item in turn is expected to exert less influence on later stages in the information processing hierarchy, such as working memory. This could also explain why the T2 representation was only initially affected (100–150 ms), as only its early appearance needs to be suppressed to stop inference with T1 processing at a higher stage. These behavioural results may be consistent with sequential working memory consolidation of targets. We found the precision of reporting T1 was unaffected by Lag, even though often during the AB only one item is reported, whereas at longer lags two items are reported. During spatial working memory tasks, where multiple items are simultaneously presented, longer lags should have a higher memory load and lead to lower precision[46]. Instead, the current results suggest that each target is consolidated into working memory before the store allows a second item to enter.

In summary, the current work adds to our understanding of the neural and behavioural basis of temporal attention. We were able to recover a neural signature for each item within an RSVP stream, something that has not been possible with conventional approaches to EEG and fMRI data. Our methodology indicated that while there is co-modulation of featural information carried by each of the targets, there is no evidence for distractor suppression in this RSVP task. We also document the existence of interactions among targets that are separated by several hundred milliseconds.

Our methodology provides a rich framework for exploring the neural bases of many psychological phenomena, including repetition blindness[47] and contingent attentional capture[48]. The current work was not designed to pinpoint the exact neural locus of the AB, but combining our approach with a technique like fMRI, which has better spatial resolution than EEG, could elucidate some of the key brain areas involved in the phenomenon. It has been suggested that feedback and feedforward processes modulate different aspects of the AB[49]. Future studies might also fruitfully combine our method with invasive recordings across multiple brain sites in animal models, to better understand the neuronal mechanisms underlying the AB effect.

## Methods

**Participants**. In Experiment 1, 22 participants (13 females, 9 males; median age 22 years; range 19–33 years) were recruited from a paid participant pool and reimbursed at AUD$20/hr. In Experiment 2, 23 participants (14 females, 9 males; median age 23 years; range 19–33 years old) were recruited from the same pool and reimbursed at the same rate. Each person provided written informed consent prior to participation and had normal or corrected-to-normal vision. The study was approved by The University of Queensland Human Research Ethics Committee and was in accordance with the Declaration of Helsinki.

**Experimental setup**. Both experiments were conducted inside a dimly illuminated room. The items were displayed on a 22-inch LED monitor (resolution 1920 × 1080 pixels, refresh rate 100 Hz) using the PsychToolbox presentation software for MATLAB[50,51]. In Experiment 1, participants were seated at a distance of approximately 45 cm from the monitor. In Experiment 2, the same viewing distance was maintained using a chinrest to minimise head motion artefacts in the EEG. At a viewing distance of 45 cm, the monitor subtended 61.18° × 36.87° (one pixel = 2.4′ × 2.4′).

**Task**. A schematic of the task is shown in Fig. 1. Supplementary Movie 1 shows two example trials. Each trial began with a central fixation point and the RSVP stream commenced after 300 ms. The stream consisted of 20 Gabors (0.71° standard deviation, ~5° diameter, 100% contrast, centred at fixation) on a mid-grey background. On each trial, the orientations of the twenty Gabors in the stream were drawn pseudo-randomly, without replacement, from integer values ranging from 0–179°. Both targets and distractors were drawn from the same random distribution, meaning there was no restriction on the relationship between targets (except they could not be identical). Note the uncorrelated nature of the targets means the design controls for possible repetition blindness effects[52], since the targets were equally likely to be similar in orientation as they were to be maximally dissimilar (i.e., orthogonal), and thus any potential orientation-specific effects would cancel out across trials.

Each item was presented for 40 ms and was separated from the next item by a blank interval of 80 ms, yielding an 8.33 Hz presentation rate. The participants' task was to reproduce the orientations of the two high-spatial-frequency Gabors (targets; 2 c/°) while ignoring the items of a low-spatial frequency (distractors; 1 c/°). Between 4 and 8 distractors, varied pseudo-randomly on each trial, were presented before the first target (T1) to minimise the development of strong temporal expectations, which can reduce the AB[40,53]. The number of distractor items between T1 and T2 defined the inter-target lag (1,2,3,5,7 in Experiment 1, and 3,7 in Experiment 2). There were 600 trials in each of the two experiments, with an equal distribution of trials across the lag conditions (120 in Experiment 1, 300 in Experiment 2), with fewer lags included in Experiment 2 to increase signal to noise for the regression-based EEG analysis. In Experiment 2, we selected Lag 3 as the test condition for the AB because it yielded a significant reduction in T2 response accuracy compared with T1 in Experiment 1, and because it has been widely used in previous studies of the AB[24,39,40,54–57].

Participants were asked to monitor the central RSVP stream until the presentation of the last Gabor, after which a response screen appeared (see Fig. 1b). The response screen consisted of a centrally presented black circle (10° diameter) and a yellow line. Participants rotated the line using a computer mouse to match the perceived orientation of the target and clicked to indicate their desired response. They were asked to reproduce the orientations of the two targets (T1, T2) in the order they were presented, and to respond as accurately as possible, with no time limit. After providing their responses, participants were shown a feedback screen which displayed their orientation judgements for T1 and T2, and the actual orientations of both targets (see Fig. 1c). The feedback was displayed for 500 ms before the next trial began, and participants were given a self-paced rest break every 40 trials. Each experiment took between 50 and 60 min to complete.

**EEG acquisition and pre-processing**. In Experiment 2, continuous EEG data were recorded using a BioSemi Active Two system (BioSemi, Amsterdam, Netherlands). The signal was digitised at 1024 Hz sampling rate with a 24-bit A/D conversion. The 64 active scalp Ag/AgCl electrodes were arranged according to the international standard 10–20 system for electrode placement[58] using a nylon head cap. As per BioSemi system design, the common mode sense and driven right leg electrodes served as the ground, and all scalp electrodes were referenced to the common mode sense during recording. Pairs of flat Ag-AgCl electro-oculographic electrodes were placed on the outside of both eyes, and above and below the left eye, to record horizontal and vertical eye movements, respectively.

Offline EEG pre-processing was performed using EEGLAB[59] in accordance with best practice procedures[60,61]. The data were initially down sampled to 512 Hz and subjected to a 0.5 Hz high-pass filter to remove slow baseline drifts. Electrical line noise was removed using the *clean_line*, and *clean_rawdata* functions in EEGLAB was used to remove bad channels (identified using Artifact Subspace Reconstruction), which were then interpolated from the neighbouring electrodes. Data were then re-referenced to the common average before being epoched into segments for each trial (−0.5 s to 3.0 s relative to the first Gabor in the RSVP). Systematic artefacts from eye blinks, movements and muscle activity were identified using semi-automated procedures in the SASICA toolbox[62] and regressed out of the signal. The data were then baseline corrected to the mean average EEG activity from 500 to 0 ms before the first Gabor in the trial.

**Behavioural analysis**. To determine how the AB affected participants' perception of targets, for each trial we found the difference between the actual target orientation and the reported orientation (i.e., the orientation error) for T1 and T2. This approach is analogous to one employed in previous work that examined whether the AB is associated with discrete or graded awareness of T2[30]. The continuous nature of the orientation responses given by participants on each trial raises the challenge of distinguishing "correct" and "incorrect" trials. For Experiment 2, we scored trials as correct when the orientation error was less than 30° from the presented orientation; trials were scored as incorrect when the orientation error was greater than 30°. As shown in Supplementary Fig. 5, this approach to scoring yielded a classic blink effect, suggesting the task captures the important behavioural features of the widely reported AB phenomenon. For each lag condition, we found the proportion of responses (in 15° bins) between −90° and +90° for the orientation errors (see Figs. 2a and 4a) and fit Gaussian functions with a constant offset (Eq. 1) using non-linear least square regression to quantify these results for each

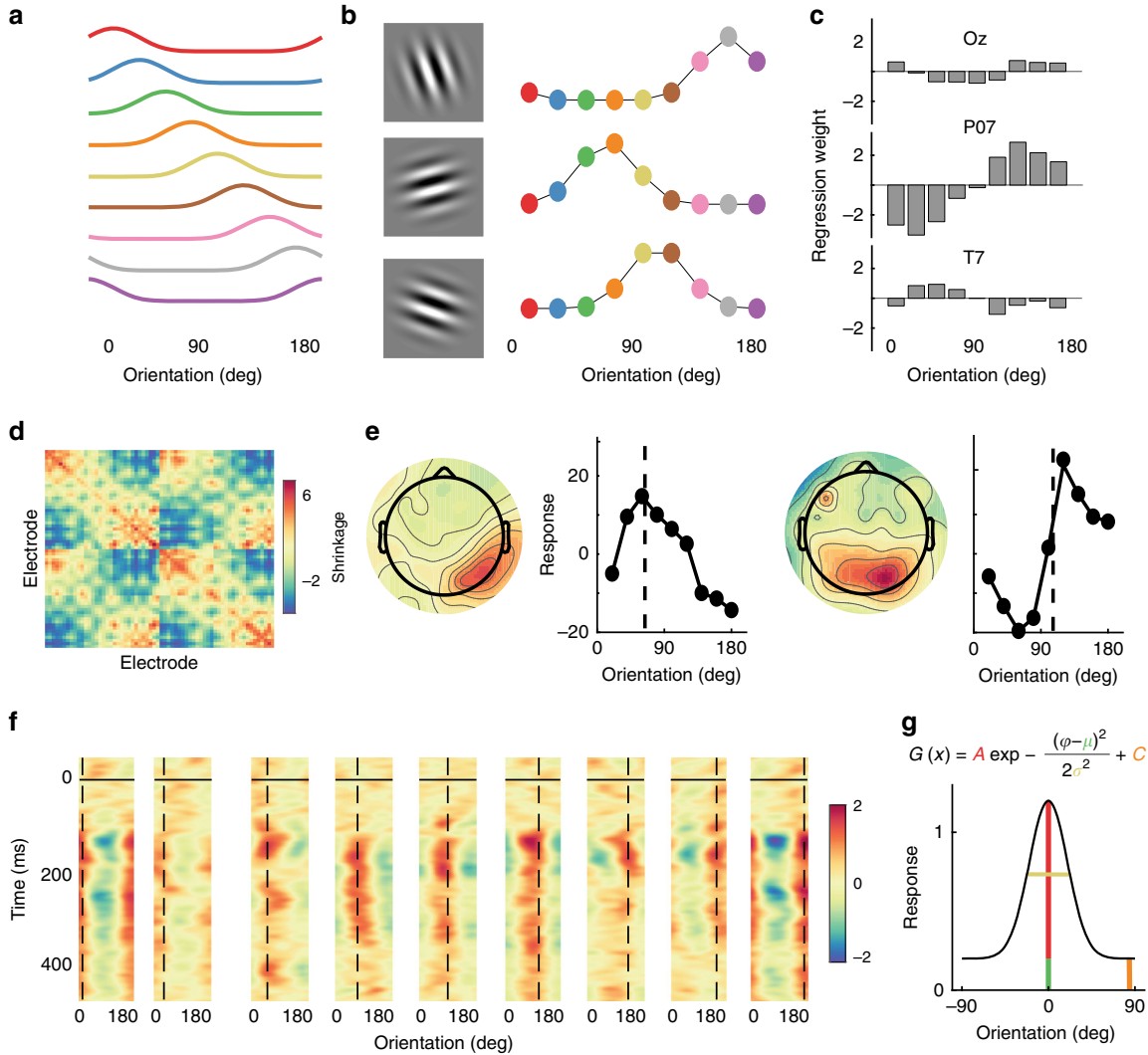

**Fig. 9 Schematic illustrating the forward encoding procedure used to estimate feature-selectivity for orientation in Experiment 2. a** A basis set of the nine channels used to model feature (orientation) selectivity. **b** The basis set was used to find the expected response (regression coefficients) for each different RSVP item in every trial, for each EEG electrode (three electrodes are shown here for a single example participant). Three trials are shown for the corresponding gratings. **c** Ordinary least squares regression was used to find regression weights for the orientation channels across trials for each EEG electrode (three electrodes are shown here for a single example participant). **d** Shrinkage matrix that the weights were divided by to perform regularisation, to account for correlated activity between electrodes. **e** The regression weights were applied to predict the presented orientation. Neural activity (headmaps) from two trials, with the channel responses for those trials. Dotted lines indicate the presented orientations. **f** Applying this procedure to each time point gives the time course of feature-(orientation) selectivity (for one participant). Trials have been binned in 20º intervals, with the dotted lines representing the presented orientation in those trials. On the y-axis, 0 ms represents the onset of the item within the RSVP stream. Feature selectivity emerged around 75 ms after stimulus presentation. **g** Modified Gaussian functions (equation) were used to quantify the tuning. The colours of the free parameters in the equation correspond to the relevant components of the tuning curve below.

participant (Figs. 2b and 4b):

$$G(x) = A \exp\left(-\frac{(x-\mu)^2}{2\sigma^2}\right) + C, \qquad (1)$$

where $A$ is the gain, reflecting the proportion of responses around the reported orientation, $\mu$ is the orientation on which the function is centred (in degrees), $\sigma$ is the standard deviation (degrees), which provides an index of the precision of participants' responses, and $C$ is a constant used to account for changes in the guessing rate. Using different bin sizes yields the same pattern of results suggesting this procedure did not bias the results (Supplementary Fig. 3). We used a Gaussian with a constant offset to characterise behavioural performance, as it captures the distribution of errors well (median $R^2 = 0.76$, SE = 0.04 in Experiment 1). This model allows the gain, width, bias and guessing rates to vary independently (Supplementary Fig. 1), unlike the function used in a previous study using a continuous report measure for the AB[30]. Most importantly, the function we implemented can also be used to characterise the forward encoding results, thus

allowing a direct comparison of the AB based upon behavioural and neural measures.

We used a regression-based approach[63] (see Figs. 2d and 4c) to determine how targets and distractors within each RSVP stream influenced behavioural responses. To do this, we aligned the orientations of both distractor and target items from 4 items prior to the appearance of the target through to 9 items after the appearance of the target to construct a regression matrix of the presented orientations. The regression matrix was converted to complex numbers (to account for circularity of orientations) using Eq. 2:

$$C_1 = \exp(1i\, C), \qquad (2)$$

where C is the regression matrix (in radians) and 1i is an imaginary unit. Standard linear regression was used to determine how the orientations of the items affected the reported orientation using Eq. 3:

$$W = \left(C_1\, C_1^T\right)^{-1} C_1^T R, \qquad (3)$$

where R is the reported orientation (in radians). This was done separately for T1 and T2 reports, with a higher regression weight indicating the item was more influential in determining the reported orientation.

To determine whether the finding that the orientations of T1 and T2 influenced the reported orientation was due to participants integrating the other target or the surrounding distractors (Fig. 3), we found the difference in orientation between the target of interest and the other item (either target or distractor) and the orientation error for each trial. This showed an orientation-tuned effect characteristic of integration. To quantitatively determine the magnitude of this effect, we fit first-derivative Gaussian functions (D1; Eq. 4) to these responses[36–38]:

$$D1(x) = A \times \frac{1}{\sigma} \times x - \mu \times \exp\left(-\frac{x-\mu^2}{2\sigma^2}\right), \qquad (4)$$

where $A$ is the gain, $\mu$ is the orientation on which the function is centred (in degrees) and $\sigma$ is the standard deviation (degrees).

**Forward encoding modelling**. Forward encoding modelling was used to recover orientation-selective responses from the pattern of EEG activity for both target and distractor items in the RSVP stream. This technique has been used previously to reconstruct colour[16], spatial[15] and orientation[19] selectivity from timeseries data acquired through fMRI. More recently, the same approach has been used to encode orientation[9,12,13] and spatial[14] information contained within MEG and EEG data, which have better temporal resolution than fMRI.

We used the orientations of the epoched data segments to construct a regression matrix with 9 regression coefficients, one for each of the orientations (Fig. 9a). This regression matrix was convolved with a tuned set of nine basis functions (half cosine functions raised to the eighth power[9,10,13], Eq. 5) centred from 0° to 160° in 20° steps.

$$F(x) = \cos(x - \mu)^8, \qquad (5)$$

where $\mu$ is the orientation on which the channel is centred, and $x$ are orientations from 0° to 180° in 1° steps.

This tuned regression matrix was used to measure orientation information either across trials or in epoched segments. This was done by solving the linear Eq. (6):

$$B_1 = WC_1, \qquad (6)$$

where $B_1$ (64 sensors $\times N$ training trials) is the electrode data for the training set, $C_1$ (9 channels $\times N$ training trials) is the tuned channel response across the training trials and W is the weight matrix for the sensors to be estimated (64 sensors $\times$ 9 channels). Following methods recently introduced for M/EEG analysis, we separately estimated the weights associated with each channel individually[13,64]. W was estimated using least square regression to solve Eq. (7):

$$W = \left(C_1 \, C_1^T\right)^{-1} C_1^T B_1. \qquad (7)$$

Following this previous work[11,13,64], we removed the correlations between sensors, as these add noise to the linear equation. To do this, we first estimated the noise correlation between electrodes and removed this component through regularisation[65,66] by dividing the weights by the shrinkage matrix. The channel response in the test set $C_2$ (9 channels $\times N$ test trials) was estimated using the weights in (7) and applied to activity in $B_2$ (64 sensors $\times N$ test trials), as per Eq. 8:

$$C_2 = \left(W \, W^T\right) W^T B_2. \qquad (8)$$

To avoid overfitting, we used cross validation (10-fold in the initial whole-trial analysis, and 20-fold when the item presentations were stacked), where X-1 of epochs were used to train the model, and this was then tested on the remaining (X) epoch. This process was repeated until all epochs had served as both test and training trials. We also repeated this procedure for each point in the epoch to determine time-resolved feature-selectivity. To re-align the trials with the exact presented orientation, we reconstructed the item representation[15] by multiplying the channel weights (9 channels $\times$ time $\times$ trial) against the basis set (180 orientations $\times$ 9 channels). This resulted in a 180 (−89° to 90°) Orientations $\times$ Trial $\times$ Time reconstruction. In order to average across trials, the orientation dimension was shifted so that 0° corresponded to the presented orientation in each trial.

For the initial encoding analysis (Fig. 5), to determine whether feature selectivity could be recovered for each RSVP item we used 20 encoding models (one for each item position in the stream) with 600 trials. We trained and tested each model across the entire 2250 ms of the trial to determine when feature selectivity emerged for that RSVP item. This analysis verified that each RSVP item could be encoded independently. We aligned all RSVP items across trials ($N = 12,000$; 600 trials by 20 items) and used a fixed encoding model for training and testing[67,68] (Figs. 6–8). This meant we trained and tested all encoding models across all items (both targets and distractors) regardless of trial type[12,13].

Aligned item reconstructions were then averaged over the relevant condition (Lag, Accuracy or item position) and smoothed using a Gaussian with a temporal kernel of 6 ms[10,13] to quantify feature selectivity. The Gaussian functions were fit, using least square regression, to quantify different parameters of feature selectivity across timepoints, as per Eq. 1, where $A$ is the gain representing the amount of feature selective activity, $\mu$ is the orientation on which the function is centred (in degrees), $\sigma$ is the width (degrees) and $C$ is a constant used to account for non-feature selective baseline shifts.

**Univariate orientation selectivity analysis**. We used a univariate selectivity analysis[10] to determine the topography associated with orientation-selective activity for targets and distractors (Fig. 8b). Data were epoched in the same manner as in the forward encoding model where EEG activity was aligned with each stream item. We separated these epochs into target and distractor presentations to determine whether these two types of stimulus were processed differently. All target presentations were used in training (1200 in total; 600 trials with two targets in each), together with a pseudo-random selection of the same number of distractor items. To determine the topography, we used a general linear model to estimate orientation selectivity for each sensor from the sine and cosine of the presentation orientation, and a constant regressor in each presentation. From the weights of the two orientation coefficients we calculated selectivity using Eq. 9:

$$A = \sqrt{B_1 \cos^2 + B_2 \sin^2}, \qquad (9)$$

$A$ was derived through permutation testing in which the design matrix was shuffled ($N = 1000$) and weights calculated. The non-permuted weights were ranked and compared with the permutation distribution, thus enabling calculation of the z-scored difference. To calculate group-level effects, cluster-based sign-flipping permutation testing ($N = 1500$) across electrodes and time was implemented in Fieldtrip[69] to determine whether the topographies differed between conditions.

**Statistics**. All statistical tests were two-sided, and Bonferroni adjustments were used to correct for multiple comparisons where noted. Non-parametric sign permutation tests[69,70] were used to determine differences in the time courses of feature selectivity (Figs. 5 and 6) between conditions. The sign of the data was randomly flipped ($N = 20,000$), with equal probability, to create a null distribution. Cluster-based permutation testing was used to correct for multiple comparisons over the timeseries, with a cluster-form threshold of $p < 0.05$ and significance threshold of $p < 0.05$.

**Reporting summary**. Further information on research design is available in the Nature Research Reporting Summary linked to this article.

## Data availability
The EEG and behavioural data for both experiments are available at: https://osf.io/f9g6h. A reporting summary for this Article is available as a Supplementary Information file.

## Code availability
The code associated with this paper is available at: https://github.com/MatthewFTang/AttentionalBlinkForwardEncoding.

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

## Acknowledgements

The authors would like to thank Brad Wyble for comments on a pre-print of this manuscript. This work was supported by the Australian Research Council (ARC) Centre of Excellence for Integrative Brain Function (CE140100007). MFT was supported by the NVIDIA corporation who donated a TITAN V GPU. EA was supported by an ARC Discovery Project (DP170100908). JTE was supported by a Discovery Grant from the Natural Sciences and Engineering Research Council (Canada). TAWV was supported by an ARC Discovery Project (DP120102313). JBM was supported by an ARC Australian Laureate Fellowship (FL110100103), an ARC Discovery Project (DP140100266), and by the Canadian Institute for Advanced Research (CIFAR).

## Author contributions

M.F.T.—conception, data gathering, data analysis, original draft, final approval. L.F.—data gathering, draft editing, final approval. E.A.—draft editing, final approval. J.T.E.—draft editing, final approval. T.A.W.V.—conception, draft editing, final approval. J.B.M.—conception, draft editing, final approval.

## Competing interests

The authors declare no competing interests.
