## [Peer Review File · Nature Communications]

Reviewers' Comments:

Reviewer #1:

Remarks to the Author:

Summary

The goal of this study was to address the neural basis of the Attentional Blink (AB). Specifically, the study employed a rapid serial visual presentation task in which participants were asked to reproduce the orientations of two target Gabors (high-frequency) among other distracting Gabors (low-frequency) with different orientations. The interval between the first target and the second target was manipulated. The behavioral results showed that when T2 was immediately presented after T1 (Lag 1 condition), subjects' response to T1 was impaired, as compared to other lag conditions. Accuracy on T2 was impaired at the short-lag conditions (e.g., Lag2 and Lag3 in Exp.1 and Lag3 in Exp.2), but not at the long-lag conditions. The Gaussian function fitting analyses showed that the reduced performance on T2 was associated with a reduction in gain (i.e., amplitude), but not the width (i.e., precision) of the T2 response. The authors argued that these patterns suggested that participants either were fully aware of the T2 or showed no awareness, supporting the discrete model of the AB. In addition, a regression analysis showed that the reported orientation of one target was biased towards the orientation of another target, and such target-to-target influence could be observed even at the long lag condition (e.g., Lag 5 condition in Exp.1). The influence of the distractors on the target mainly came from the item which was presented immediately after that target.

In Experiment 2, the authors recruited a new set of subjects doing the same RSVP task while and recording the EEG data when subjects did the task. Using a multivariate pattern analysis approach (i.e., forward encoding model), they reconstructed the orientation-selective response, and examined how the representation of the target was affected by that of another target, as well as those of distractors. The EEG results showed that, within a narrow range of time window (i.e., 100-150ms after the onset of the target), the orientation representation of T2 was boosted when the target was correctly reproduced, whereas the representation was suppressed when the T2 target was missed. This effect was only observed for Lag3 condition, but not for lag7.

Overall Impression:

This is an interesting study, with a novel multivariate approach to examine the neural basis of the AB. The design seems sound, and the analyses are mostly justified. The study provides some important results and may have some significant contribution to both theoretical and neural basis of the AB, though I have concerns that mostly focus on the relationship between behavior and brain data, and their choices (and executions) of analyses.

Major Concerns:

The behavioral results suggest a discrete model of the AB, but in the EEG analyses, the author split the trials into correct and incorrect based on an arbitrary partition (e.g., within 30° of the presented orientation) and the EEG data is more consistent with a graded model of the AB. To provide a strong link between the neural data and behavioral data, they should examine if there is a correlation between the subject-reported angular error and the neural orientation selectivity response amplitude.

The authors should include a confusion error in their analysis: Figure 2C shows evidence of T1 and T2 confusions at Lag2 and Lag3 conditions (i.e., substitution errors). Also, the study used a 4-parameter Gaussian function, which is different from the method (Asplund et al., 2014) that authors just cited, to fit the behavioral data but did not justify this approach.

In Figure 6A, the blue and red lines in the right panel for Lag7 correspond to incorrect and correct trials, but there were no differences between their time course. That is, the correct and incorrect T2 trials yield similar EEG profiles, casting doubt that differences in the timecourses between correct and incorrect trials at lag 3 bear any relationship to the behavior.

Relatedly, the Figure 6 shows suppression for incorrect trials at Lag3 condition, but not complete suppression (e.g., as compared to the zero). Also, in the same figure for Lag3, the authors did not provide an explanation why the orientation selectivity was going back up for the incorrect trials (blue line) after the critical period marked by the black bar?

In Figure 2, there might be an effect of Lag on width for T2, but they did not include lag as a factor in statistical analyses. They should do so instead of looking at the difference between T1 and T2.

In Exp.2, they included only Lag3 and Lag7 conditions for power reasons. However, in Exp.1, it seems that the Lag2 condition had a bigger effect size than Lag3 in term of amplitude measure (Figure 2B). Also, Lag2 condition for width in Exp.1 suggests that it far way more power than the lag3 condition. So, to me, the analysis should be run on either lag2 or lag 2 and 3 combined rather than just lag 3.

Minor comments:

For Expt 1, the authors should explicitly note that unlike in typical AB studies, in which T2 performance is assessed for T1 correct trials only, in their task they cannot easily select the correct T1 trials given the orientation reproduction task on T1. (Could this issue bias the results on T2?)

The behavioral result in Exp. 1 showed that at lag 1 condition the response to T1 was strongly influenced by T2, and in the Methods, it is not clear whether there was any restriction on the difference in orientations of T1 and T2. If the orientations of T1 and T2 are similar and close to each other at the lag 1 condition, such impairment could be due to the repetition blindness (Johnston, Hochhaus and Ruthruff, 2002, *Journal of Experimental Psychology: Human Perception & Performance*) which also refers to participants' impaired ability in reporting both repeated targets in an RSVP task.

In fig. 7A, distractor orientation info is as good at lag 2 (trough of the AB) than it is at lag 7, which is intriguing.

Reviewer #2:

Remarks to the Author:

Tang et al sought to understand how neural representations of targets in the traditional attentional blink task impact one another. To do so, they implemented a continuous whole report version of the attentional blink task using oriented gratings and sampled multiple T1/T2 lags, with targets defined as changes in spatial frequency, and reports requiring precise responses to T1 and T2 orientations. By carefully examining behavioral reports, they determined that target (T1) reports are biased towards immediately-following distractors, and towards each other (T1/T2), but not towards intervening distractors. Then, in a second experiment, they recorded EEG while participants performed the task. By implementing a multivariate analysis that allowed them to assay the information content of scalp potential patterns at each timepoint in the trial (and to each successive oriented grating stimulus), they could determine the neural 'fate' of each stimulus representation, sorted based on whether it was T1, T2, or a distractor. Doing so illustrated that the presence of an AB (error in T2 report) was associated with a suppression in the amplitude of the model-based EEG stimulus representation compared to intervening distractors (and, accordingly, correct T2 reports, even at short lags, were associated with intact neural representations). Because there was no evidence that distractor representations were enhanced or suppressed, the authors could rule out a class of proposed AB mechanisms which involve T1-induced changes in distractor processing. Instead, the authors suggest a different mechanism, whereby the ability to re-engage with T2 (and T2 alone; rather than necessarily including distractors) determines the fate of the neural representation, and thus whether

an AB occurs.

I think this is a really nice contribution that is elegantly presented, and would be of substantial interest to the broader neuroscience community. What sets this study apart, to me, is that it's the first time (in my view) that decoding neural representations at the timescale of EEG was really necessary to answer the research question. I would suggest the authors expand the presentation of the Methods, consider some alternative statistical tests, and adjust a few figures, but these are small suggestions that would only enhance an already impressive report. I list several comments below the authors may wish to address.

1. Throughout, the authors use what they call a "forward encoding" approach to recover orientation-selective neural representations. This is very similar to the alternatively-named "inverted encoding model" analysis procedure that's recently come under fire by Gardner & Liu (see Liu et al, 2018, *J Neuro*; Gardner & Liu, 2019 *eNeuro*; as well as commentaries by Sprague et al in 2018, *eNeuro* and 2019, *bioRxiv*). While I don't think the Liu & Gardner concerns are particularly strong with regard to the general efficacy of the technique, they make several important points about the proper presentation of results. As such, I have a few recommendations for the authors that would minimize pushback from that community:

a. be careful about words like "orientation selectivity" – instead try to use phrases like "orientation information" or "feature-selective stimulus reconstruction", etc, that make extraordinarily clear the analyses are not with reference to single-neuron tuning/selectivity. I think the authors do a great job already, but I did see a few examples (e.g., lines 35, 94, 413; 516-522) where this could be made more precise. The Sprague, Boynton & Serences commentary makes several recommendations about nomenclature the authors may be interested in considering, though of course the authors can and should use the language they're most comfortable with

b. more clearly describe and illustrate, graphically, all aspects of the encoding model-based analysis procedure. It seems as though the authors are more or less adopting the modified implementation of forward models presented by Kok et al (2017), but the text in the methods is quite sparse. In the interest of full transparency, I recommend the authors vastly expand on this, and include additional details like specific basis function equations, etc. One potential issue w these model-based analysis methods is that the results necessarily depend on the model used for analysis, and so very clearly and transparently reporting all aspects of the model is essential. I'd recommend even adding a figure illustrating this, if possible. (minor: Eq 5 uses superscripts instead of subscripts for B2, C2)

c. There are important considerations regarding how to compare reconstructions when different estimated models are used for reconstruction (highlighted in Sprague et al, 2018; their discussion of "fixed" encoding models). This seems particularly relevant to this report, especially for the analyses presented in Figs. 6 and 7A, where parameters of fit Gaussians to encoding models are directly compared across target presentation conditions (T1, T2, distractor; correct/incorrect). Can the authors describe these model estimation procedures in more detail? For the training set, were only distractor/T1/T2 epochs chosen? Or all of them? I don't think there's necessarily a most 'right' way to do this – but there should be a bit more detail in the manuscript to help readers understand the authors' procedures (see also code sharing recommendations, below).

d. Amplitude vs fidelity – the use of "amplitude" seems pretty straightforward (gain of Gaussian), but "fidelity" should be more clearly defined – it seems like the authors are using this to describe the "width" of the Gaussian (line 182)? If so, perhaps something like FWHM or width would be more appropriate? (e.g., you could, in principle, have a low amplitude but high fidelity, or low fidelity but high amplitude result, both of which feel counterintuitive with that language).

2. Data & code accessibility/availability statement – the authors should clearly describe, in the manuscript, how they plan to make data and analysis code available to the community. I recommend using OSF.io and github.com to ensure persistent availability. Additionally, sharing code will improve the transparency of the methods. I see that there's a software submission checklist included, but I can't find a reference to this in the manuscript itself. Indeed, in the reporting summary, the authors mention that code/data are available upon "reasonable request" – I encourage the authors to relax

this requirement and post the code/data somewhere publicly accessible. But, in any case, this should be in the manuscript itself.

3. Statistics – In several places (e.g., lines 380-381), the authors show significant differences in one comparison (Lag 3) but not another (Lag 7), but do not report the test for interaction – the appropriate test would be a 2-way ANOVA (factors of Lag and target #). This also comes up earlier – Lines 297-307 and Fig 4, and in Fig. 6A (interaction between lag and accuracy). Additionally, there doesn't seem to be much detail included about the permutation procedures used for evaluating encoding model success (Fig. 5) and differences between conditions (Fig. 6). The authors refer to a "two-tailed cluster-permutation" test (line, e.g., 363), but I'm not sure what is being permuted? Are trial labels during training shuffled for each subj, then results from such a null model compared against the real results? How does the analysis which looks for differences between conditions (Fig. 6) compared to that which looks at significant information (Fig. 5)? Finally, in Figure 3, it seems like Bonferroni correction for multiple comparisons was performed (Line 268) – what about for the similar tests shown for Fig. 2B?

4. Regression analysis – I'm not sure I understand this procedure, largely because it is very sparsely described in the Methods (Lines 640-646). What goes into the analysis? A circular variable of raw report angle? If so, how is the circularity of the variable accounted for? While I know most readers understand typical linear regression, in this case it can be helpful to very clearly write out what data is used to predict what, and how the authors treat the circular variables.

5. Role of working memory – the authors briefly mention working memory (line 511), but I'm more interested in how the maintenance of information over a relatively extended delay may have impacted behavioral performance. Especially because, in some cases (like where T2 is poorly encoded because it's missed due to the AB), the working memory load might actually be different – subjects are storing only one orientation rather than two. This isn't a criticism of the study by any means – I'm just curious if the authors could speak to how their results may also reflect changes in working memory load across conditions. I think their results show something neat, and potentially unexpected in the working memory literature – the shift from Load 1 to Load 2 does not seem to alter precision, instead only guessing (Fig. 4B).

Reviewer #3:

Remarks to the Author:

Summary:

Tang et al. leverage forward modeling of EEG activity to examine stimulus-specific representations during the attentional blink (AB). Participants viewed rapid streams of oriented stimuli and recalled the orientations of two targets (T1 and T2) defined by a unique spatial frequency. Behavioral measures revealed impaired processing of T2 when it was presented 100-400 ms after T1 – the standard AB effect – while reports of each target's orientation exhibited a significant bias towards the other target's orientation (i.e., reports of T1 were biased toward the orientation of T2 and vice versa). Forward modeling of concurrent EEG signals revealed robust orientation-specific representations of targets and distractors over periods spanning ~75-250 ms after stimulus onset. However, representations of T2 were significantly attenuated when participants misreported T2 by $> \pm 30^\circ$. Supplementary analyses revealed stronger representations of T1 and T2 relative to distractors during "correct" trials, but not during "incorrect" trials. The authors conclude that impaired T2 processing during the AB reflects prolonged processing of T1 rather than inadvertent processing of distractors.

Evaluation:

This is an interesting study that makes clever use of multivariate analyses to make a provocative

point. The novelty here is the use of an encoding model to directly track changes in the fidelity of target and distractor representations during an RSVP task, which could yield new and important insights into the mechanisms responsible for the AB. Overall I found the results compelling but have some concerns about whether the conclusions are justified. The authors should be able to address these in a revision. For clarity, I've enumerated my questions below.

Signed,

Edward Ester

1. Re: modeling the behavioral data, typically report errors are modeled with parametric models assuming that the observed data are a mixture of different trial outcomes (e.g., correctly reporting the to-be-recalled item with precision k vs. guessing; Zhang & Luck, 2008; Bays et al., 2009; Bays, 2016). The author's model captures the spirit of that approach, with the center, width, and baseline parameters capturing bias, imprecision, and guessing rates. But it's not clear what aspect of memory "amplitude" refers to, nor how it should be interpreted. Mathematically the parameter is a multiplicative scalar that controls the peak of the response relative to baseline, which implies a kind of fidelity or signal-to-noise ratio. The problem with that interpretation is that the authors are fitting a curve to a relative frequency distribution that sums to 1, and changing the baseline parameter (guessing rate) necessarily changes the amplitude (and/or width) parameter. So, what's really going on here? Is it the fidelity of the representation or the likelihood of missing T2 and guessing that's changing across lags? Both outcomes would be consistent with global workspace theory (p. 12) but would motivate very different interpretations of the data and how they map onto the fidelity of reconstructed neural representations.
2. Related: using the authors' approach the parameter estimates returned by the model will vary to some extent based on how the data are binned (e.g., Fig 2A). A likelihood-based approach that leverages the full set of report errors from each participant could yield a more stable estimate. It'd be nice to see the outcome of such an analysis, if only to show that both approaches yield comparable results.
3. The authors fit model-based reconstructions of stimulus orientation with the same function used to characterize behavioral performance and use the amplitude parameter as an index of orientation selectivity. That's fine, but it'd be helpful to know if there are differences in reconstruction bandwidth or bias across lags. If so, then the authors' selectivity index might be missing important information that could inform their conclusions. Showing that there's a difference in amplitude but not bandwidth during correct vs. incorrect trials addresses this to some extent, but it'd be helpful to see the same analysis applied to the full data set.
4. The authors compare the relative strengths of reconstructed target and distractor representations in an attempt to adjudicate between T1-based and distractor-based accounts of the AB (p. 24-25). During correct trials, they report stronger representations of T1 and T2 relative to distractors. During incorrect trials, they report no difference in representation strength between T1 and distractors and reduced representation strength for T2 relative to distractors. That implies reduced processing of T1 and T2 during correct vs. incorrect trials, but it doesn't really tell us anything about whether the AB reflects prolonged processing of T1 or inadvertent processing of distractors. To test the T1 processing model the authors could compare T2 selectivity as a function of T1 selectivity: T2 selectivity should be reduced (and report errors higher) when T1 selectivity is high vs. low during T2 presentation. To test the distractor model the authors could compare selectivity for distractors presented before T1 to distractors presented during T1 and T2: distractors presented during the AB should have a higher selectivity during the AB period relative to distractors presented before or after both targets.
5. The authors' findings are broadly consistent with an "early-stage" view of the AB where impairments in T2 reports reflect impaired perceptual processing. Yet other EEG studies cited by the

authors (e.g., Vogel, Luck, Shapiro) imply that stimuli can be accessed but not reported during the AB. Can the authors square their findings with these earlier reports? Some additional discussion of this point would be helpful.

6. The authors define correct trials as $\leq 30^\circ$ report error for T2 given that T1 was correctly reported (presumably "T1 correct vs. incorrect" was defined based on the same criterion? I couldn't find this information in the manuscript). I understand that this criterion was chosen to yield and approximately equal number of correct and incorrect trials (related: it'd be helpful if the authors stated average \pm SEM correct/incorrect trials across participants somewhere in the manuscript). However, $\pm 30^\circ$ seems like a liberal threshold: if report errors are scored from -90 to 89, then the average absolute recall error for a participant who randomly guesses on each trial is $\pm 45^\circ$. Do the results change appreciably if a more stringent criterion is adopted?

Response to reviewers

We are pleased with the high level of enthusiasm expressed by the reviewers for our work and would like to thank them for their thoughtful evaluations.

We believe we have significantly improved the manuscript after carefully considering the comments and revising the text to address each of the issues raised (outlined in detail below). In addition, following the reviewers' requests and the laudable policy of *Nature Communications*, we have made all of the data and code available through open-access repositories.

In the text that follows, reviewer comments are shown **in bold** and our responses to them immediately below in normal font. For convenience, we have reproduced major changes in the manuscript text, verbatim, **in red font**.

Reviewer 1

Summary

This is an interesting study, with a novel multivariate approach to examine the neural basis of the AB. The design seems sound, and the analyses are mostly justified. The study provides some important results and may have some significant contribution to both theoretical and neural basis of the AB, though I have concerns that mostly focus on the relationship between behavior and brain data, and their choices (and executions) of analyses.

Major Concerns:

1. The behavioral results suggest a discrete model of the AB, but in the EEG analyses, the authors split the trials into correct and incorrect based on an arbitrary partition (e.g., within 30° of the presented orientation) and the EEG data is more consistent with a graded model of the AB. To provide a strong link between the neural data and behavioral data, they should examine if there is a correlation between the subject-reported angular error and the neural orientation selectivity response amplitude.

We thank the reviewer for this comment, which is similar to a question raised by Reviewer 3, point 6. In the original manuscript, we included a supplementary figure showing that the EEG results were not due to an arbitrary split into correct and incorrect trials. For this analysis, we showed that orientation selectivity in the EEG data was systemically related to the magnitude of the behavioural orientation error. Selectivity was greatest on trials with small errors, and smallest on trials with large errors. This pattern was supported by a one-way ANOVA, which showed a significant effect of error magnitude on the gain of the neural representation. Note that we did not undertake a correlational analysis because the forward encoding results are too variable on a trial-by-trial basis to fit Gaussians precisely. Instead we sorted the data into 15° orientation-error bins, which produces much more reliable encoding results. As this point was raised by two reviewers, we have now included

this analysis in the main text and expanded on the interpretation of the results in the revision.

Page 24, Line 450

The reduction in T2 selectivity for incorrect trials at Lag 3 was not driven by an arbitrary split of trials into correct and incorrect categories. To verify this, we sorted the evoked T2 forward encoding results by the amount of orientation error (in 15° error bins to allow sufficient signal-to-noise ratios for fitting). There was significantly greater feature selectivity when the orientation error was small, and this selectivity decreased gradually with larger errors (one-way ANOVA, $F(1,22)=2.76$, $p=0.02$, $\eta_p^2=0.11$; Figure 7).

Figure 7. Gain of feature-selective information for T2 items presented at Lag 3 in Experiment 2, plotted as a function of reporting error. Forward encoding results were averaged across early time points (100-150ms), and were binned by the absolute difference between the presented and reported orientations (in 15° increments). Each bin is displayed as the starting value (e.g., 0° incorporates errors from 0° to 15°). Gaussians were fitted to quantify selectivity with the gain parameter shown here. Feature selectivity was highest when participants reported the orientation to within 30° of the presented orientation, and declined significantly with larger reporting errors. Error bars indicate ± 1 standard error of the mean.

2. The authors should include a confusion error in their analysis: Figure 2C shows evidence of T1 and T2 confusions (i.e., substitution errors) at Lags 2 and 3.

The reviewer is correct that there were some substitution errors present at early Lags, which we labelled as “switching errors” in the original manuscript. We have included a confusion error analysis as Supplementary Figure 4 in the revised

manuscript. We thank the reviewer for suggesting this analysis, as it helps to highlight the key points of our behavioural results.

Page 47, Line 927

Supplementary Figure 4. Heatmaps of reported versus presented orientations across all participants in Experiment 1. These are analogous to confusion matrices for a continuous report task. Warmer colours indicate a greater proportion of responses. Note that participants were asked to report the targets in their presented order. Each row is a lag and each column shows a different comparison. Panels in the leftmost column show presented-T1 orientation against reported-T1 orientation. For Lags 2-7, there is a strong correspondence between presented and reported orientations, confirming that participants accurately reported T1 targets. The next column to the right shows the outcome of the same analysis, but for T2 targets. For these items, there was a strong correspondence between presented and reported orientations at Lags 1, 5 and 7, which decreases (i.e., more random) for items at Lags 2 and 3. The next column shows T1 switching, where presented-T1 orientation is plotted against reported-T2 orientation. The rightmost column shows T2 switching, where presented-T2 orientation is plotted against reported-T2 orientation. Clear switching is evident only at Lag 1, where the orientation of the item presented at T2 is reported as the orientation of T1.

3. Also, the study used a 4-parameter Gaussian function, which is different from the method (Asplund et al., 2014) that authors just cited, to fit the behavioral data but did not justify this approach.

We thank the reviewer for highlighting this omission. We have now included a justification for our choice of parameters for quantifying selectivity in the revised manuscript. Supplementary Figure 1 shows the benefits of using this function over that used by Asplund et al.

$$G(x) = A \exp\left(-\frac{(\varphi - \mu)^2}{2\sigma^2}\right) + C$$

A = Gain
 σ = Width
 C = Baseline

μ = Centre orientation
 set to 90 deg

Supplementary Figure 1. Examples of the four-parameter Gaussians used to quantify behavioural orientation errors and forward encoding representations for orientation from EEG activity. The figure shows the effect of different parameter values on the shape of the resulting function. Each row has a different gain value, and each column has a different width parameter. Within each panel, the baseline value changes. The width parameter shows the precision (of either the behavioural responses or neural representations). The baseline parameter captures non-selective responses that are unrelated to the target. For the behavioural analysis, this reflects random guessing which would be distributed equally across all orientations; for the EEG analysis, it reflects overall, non-feature selective activity from the orientation encoding. For all panels, the centre orientation of the Gaussian is set to 90°. The figure highlights the independence of the parameters of the Gaussians. For instance, looking at the panels across a given row (where width varies, but gain is fixed) reveals that curves with the same baseline value have peaks at the same height. Inspecting any one panel shows that the baseline and gain parameters are independent, with the differences between the peaks of the curves being equal regardless of the baseline.

4. In Figure 6A, the blue and red lines in the right panel for Lag7 correspond to incorrect and correct trials, but there were no differences between their time course. That is, the correct and incorrect T2 trials yield similar EEG profiles,

casting doubt that differences in the timecourses between correct and incorrect trials at lag 3 bear any relationship to the behavior.

The reviewer is correct that in Figure 6A there were no differences in neural representation of orientation between correct and incorrect trials at Lag 7, whereas there were differences between correct and incorrect trials at Lag 3. This finding matches the behavioural results, which showed T2 accuracy was severely degraded at Lag 2 compared with T1 accuracy, but this difference was not apparent at Lag 7. This pattern is entirely consistent with the AB effect as found here and in many previous investigations (see Dux and Marois, 2009 for review). Theories of the AB predict that behavioural and neural differences should emerge when successive targets are presented with a short delay between them (e.g., at Lag 3), but not at longer delays (e.g., at Lag 7). At Lag 3, errors primarily arise due to a dual-task cost (i.e. the AB), which we believe causes suppression of the T2 representation. At Lag 7, T2 accuracy is similar to T1 accuracy, suggesting that errors are now caused by general task engagement (mind wandering, etc), and are therefore not related to the AB. If we had found differences between correct and incorrect trials at both Lags, it would suggest overall task demands rather than the AB affects the neural representation. We have highlighted this in the revised manuscript.

Page 22, Line 417

There were no significant differences in the orientation-selective response between correct and incorrect trials for T2 at Lag 7, suggesting the suppression is caused by the AB rather than general target detection. This was expected because the AB typically lasts less than 500 ms, and is consistent with the current behavioural results showing an AB at Lag 3 but not at Lag 7.

5. Relatedly, the Figure 6 shows suppression for incorrect trials at Lag3 condition, but not complete suppression (e.g., as compared to the zero).

The reviewer is correct that we did not observe complete suppression of T2 at Lag 3 for incorrect trials. We were deliberately conservative in our original description of this result, since our statistical analyses focused on differences between matched conditions (i.e. when the stimuli were identical while behaviour differed). We now describe the result as follows: Correct trials were associated with significantly higher gain than incorrect trials (Figure 6), and incorrect trials were associated with significantly lower gain than distractors (original Figure 7, now Figure 8). We have re-phrased some of these points in the revision to avoid confusion.

Page 22, Line 415

For T2, however, incorrect trials resulted in a significantly decreased feature-selective response (cluster $p=0.02$) relative to correct trials shortly after each item appeared (100 to 150 ms) at Lag 3, although the response was not completely suppressed.

Page 27, Line 511

Note that all distractors and targets have gain values significantly above 0 arbitrary units (a.u.), indicating robust feature-selectivity.

6. Also, in the same figure for Lag3, the authors did not provide an explanation why the orientation selectivity was going back up for the incorrect trials (blue line) after the critical period marked by the black bar?

Our preferred explanation is that T2 is only affected from 100 to 150 ms after presentation because only its early sensory representation is suppressed when T1 processing is ongoing. This might occur because suppression inhibits the sensory representation from being passed to later cortical areas where working memory consolidation occurs. We have included a discussion of this in the revision.

Page 30, Line 587

Suppression of the T2 representation occurred 100-150 ms after the target appeared, suggesting inhibition by ongoing processing of T1. This fits well with previous work showing that the AB is associated with a reduced late-stage response, as indicated by an ERP component associated with working memory consolidation^{8,46}. Taken together with the current results, it seems that suppressing the initial T2 representation is likely to inhibit information from being passed to later cortical stages associated with working memory consolidation. As this information does not reach higher cortical stages, it results in a reduced working memory ERP component during the AB. This could also explain why the T2 representation was only initially affected (100-150 ms), as only its early appearance needs to be suppressed to stop inference with T1 processing at a higher stage.

7. In Figure 2, there might be an effect of Lag on width for T2, but they did not include lag as a factor in statistical analyses. They should do so instead of looking at the difference between T1 and T2.

Thank you for this suggestion. We have included this analysis in the revision. In summary, the ANOVA revealed no significant effects of the width parameter.

Page 11, Line 194

The accuracy reduction for T2 at Lags 2 and 3 was primarily linked to a reduction in gain. A 2 (Target; T1,T2) x 5 (Lag; 1,2,3,5,7) within-subjects ANOVA showed the gain parameter was affected by Target ($F(1,21)=10.00$, $p=0.005$, $\eta_p^2=0.32$) and Lag ($F(4,84)=11.66$, $p<0.0001$, $\eta_p^2=0.36$), and the interaction between these factors ($F(4,84)=7.10$, $p<0.0001$, $\eta_p^2=0.25$). Critically for our first theoretical question, the spread (width) of orientation errors was unaffected by the factors of Target ($F(1,21)=0.10$, $p=0.76$, $\eta_p^2=0.005$) or Lag ($F(4,84)=0.55$, $p=0.70$, $\eta_p^2=0.03$), or by the interaction between these factors ($F(4,84)=0.19$, $p=0.94$, $\eta_p^2=0.01$). The baseline parameter, which reflects guessing of random orientations, was also significantly affected by the factors of Target ($F(1,21)=12.72$, $p=0.002$, $\eta_p^2=0.38$) and Lag ($F(4,84)=4.82$, $p=0.002$, $\eta_p^2=0.19$), and there was a significant interaction between them ($F(4,84)=5.04$, $p=0.001$, $\eta_p^2=0.19$). These same effects were also evident when the data were not normalized (Supplementary Figure 2), and with a wide range of parameters to specify the orientation errors (Supplementary Figure 3).

8. In Exp.2, they included only Lag3 and Lag7 conditions for power reasons. However, in Exp.1, it seems that the Lag2 condition had a bigger effect size than Lag3 in term of amplitude measure (Figure 2B). Also, Lag2 condition for width in Exp.1 suggests that it far way more power than the lag3 condition. So, to me, the analysis should be run on either lag2 or lag 2 and 3 combined rather than just lag 3.

We would like to clarify that there were more T1-T2 Lag conditions in Experiment 1 (behaviour only) than in Experiment 2 (behaviour plus EEG). In Experiment 1, there were five Lags (1, 2, 3, 5 and 7) with 120 trials for each condition. In Experiment 2, by contrast, we included Lags 3 and 7 only, in order to increase the number of trials per condition (300 in each) and thus to maximise statistical power for the EEG encoding analyses. As the reviewer recognises, the effects we observed in the EEG data at Lag 3 would likely be even stronger at Lag 2 (as suggested by the behavioural results from Experiment 1) if we had included this lag in Experiment 2. The fact that we nevertheless observed statistically significant effects at Lag 3 in Experiment 2 indicates that the effect was still sufficiently strong. We have revised relevant sections of the Methods and Results to ensure that readers appreciate the difference in the number of Lag conditions included in Experiments 1 and 2.

Page 16, Line 313

The method was identical in all respects, except that we now included targets only at Lags 3 and 7 (i.e., a single target inside and outside the AB, respectively) to increase the within-subject power for the EEG analyses.

Page 34, Line 655

There were 600 trials in each of the two experiments, with an equal distribution of trials across the lag conditions (120 in Experiment 1, 300 in Experiment 2), with fewer lags included in Experiment 2 to increase signal-to-noise for the regression-based EEG analysis.

Minor comments:

For Expt 1, the authors should explicitly note that unlike in typical AB studies, in which T2 performance is assessed for T1 correct trials only, in their task they cannot easily select the correct T1 trials given the orientation reproduction task on T1. (Could this issue bias the results on T2?)

The reviewer is correct about the difference between conventional analyses of AB performance for categorical stimuli (e.g., letters, digits) and our procedure for analysing orientation judgements as a continuous variable. We have included a discussion of this point in the revised manuscript. However, this particular aspect of our analytic approach is unlikely to have unduly biased the results. Many previous studies have reported analyses of T2 accuracy regardless of T1 detection (e.g. Martin, Enns, Shapiro, 2011, *PBR*; Goodbourn, et al, 2016; *Psych Sci*; Lasaponara, et al, 2015, *Cortex*). These studies reveal that the pattern of results is generally similar to that observed when data are analysed exclusively for T1 correct trials. Note also that in our analyses for Experiment 2, we only included trials in which participants responded within 30 degrees of the correct T1 orientation. Using this

criterion, we observed a classic AB effect (similar to the gain parameter from the modelling results), such that the factors of Lag and Target significantly interacted to affect accuracy ($p < .05$), with T2 accuracy at Lag 3 being lower than T1 accuracy. This finding demonstrates that an RSVP task in which participants reproduce the orientations of target Gabors can replicate the hallmarks of the AB for categorical stimuli like letters and digits. These considerations have now been elaborated in the revised manuscript, and a further figure (Supplementary Figure 5) has been added.

Page 21, line 397

To determine the effect of the AB on orientation-selectivity, we separated the forward encoding results by target (T1,T2) and T2 accuracy (correct, incorrect). For the purposes of the analyses, trials were scored as correct if the reported orientation was within ± 30 degrees of the presented orientation, a criterion which yielded roughly equal correct and incorrect trials at Lag 3. In line with the AB literature, for all the EEG analyses we only included trials where participants correctly identified T1. Applying these criteria yielded the classic AB effect (Supplementary Figure 5). A 2 (Lag; 3,7) x 2 (Target; T1,T2) within-subjects ANOVA applied to these scores revealed significant main effects of Lag ($F(1,22)=199.52$, $p < 0.0001$, $\eta_p^2=0.90$) and Target ($F(1,22)=8.58$, $p=0.008$, $\eta_p^2=0.28$), and a significant interaction ($F(1,22)=9.64$, $p=0.005$, $\eta_p^2=0.31$). Follow-up t-tests showed that Lag 3 accuracy was significantly lower than Lag 7 accuracy for T2 items ($t(22)=3.15$, Bonferroni $p=0.01$, $d=0.66$) but not for T1 items ($t(22)=1.97$, Bonferroni $p=0.12$, $d=0.41$).

Page 35, Line 698

The continuous nature of the orientation responses given by participants on each trial raises the challenge of distinguishing “correct” and “incorrect” trials. For Experiment 2, we scored trials as correct when the orientation error was less than 30° from the presented orientation; trials were scored as incorrect when the orientation error was greater than 30° . As shown in Supplementary Figure 5, this approach to scoring yielded a classic blink effect, suggesting the tasks captures the important behavioural features of the widely-reported AB phenomenon.

The behavioral result in Exp. 1 showed that at lag 1 condition the response to T1 was strongly influenced by T2, and in the Methods, it is not clear whether there was any restriction on the difference in orientations of T1 and T2. If the orientations of T1 and T2 are similar and close to each other at the lag 1 condition, such impairment could be due to the repetition blindness (Johnston, Hochhaus and Ruthruff, 2002, Journal of Experimental Psychology: Human Perception & Performance) which also refers to participants’ impaired ability in reporting both repeated targets in an RSVP task.

The orientations of items in the RSVP streams were uncorrelated, and there was no restriction on the orientations of the T1 and T2 targets. Thus, across trials, there was no systematic relationship between targets, and thus any incidental correlation in orientations within a specific trial would sum to zero across the experiment. In this way, our design controlled for any unwanted biases (e.g., due to repetition blindness). Furthermore, repetition blindness does not occur when targets are separated by multiple items, as was the case for all but the Lag 1 conditions in our

RSVP task. For Lag 1 trials (in Experiment 1), a repetition blindness-like effect may have contributed to the results when targets had similar orientations. Our behavioural paradigm could be redesigned to investigate repetition blindness effects, as we have done in the current analysis to examine target integration. We have noted this in the Discussion section of the revised manuscript.

Page 32, Line 613

Our novel methodology provides a rich framework for exploring the neural bases of many psychological phenomena, including repetition blindness⁴⁸ and contingent attentional capture⁴⁹.

Page 33, Line 645

Both targets and distractors were drawn from the same random distribution, meaning there was no restriction on the relationship between targets.

In fig. 7A, distractor orientation info is as good at lag 2 (trough of the AB) than it is at lag 7, which is intriguing.

Yes, this is correct. We believe that distractor information is comparable for items +2 and +7 because only target information is suppressed by the central mechanism that gives rise to the AB. This result supports our conclusion that targets, but not distractors, are inhibited during the AB. We have emphasized this point in the revised manuscript.

Page 26, Line 496

To directly test the distractor model of the AB, we compared distractor representations before T1 with distractor representations during the AB (i.e., between T1 and T2). The account predicts that distractors presented during the AB should elicit a stronger neural representation as they are likely to be incorrectly selected as targets. Instead, we found that distractors were represented similarly before and during the AB for both correct trials ($t(22)=0.85$, $p=0.40$, $d=0.18$) and incorrect trials ($t(22)=1.83$, $p=0.08$, $d=0.38$).

Reviewer 2

1. Throughout, the authors use what they call a “forward encoding” approach to recover orientation-selective neural representations. This is very similar to the alternatively-named “inverted encoding model” analysis procedure that’s recently come under fire by Gardner & Liu (see Liu et al, 2018, J Neuro; Gardner & Liu, 2019 eNeuro; as well as commentaries by Sprague et al in 2018, eNeuro and 2019, bioRxiv). While I don’t think the Liu & Gardner concerns are particularly strong with regard to the general efficacy of the technique, they make several important points about the proper presentation of results. As such, I have a few recommendations for the authors that would minimize pushback from that community:

a. be careful about words like “orientation selectivity” – instead try to use phrases like “orientation information” or “feature-selective stimulus

reconstruction”, etc, that make extraordinarily clear the analyses are not with reference to single-neuron tuning/selectivity. I think the authors do a great job already, but I did see a few examples (e.g., lines 35, 94, 413; 516-522) where this could be made more precise. The Sprague, Boynton & Serences commentary makes several recommendations about nomenclature the authors may be interested in considering, though of course the authors can and should use the language they’re most comfortable with

Thank you for highlighting this point. We have altered our terminology along the lines suggested throughout the revised manuscript.

b. more clearly describe and illustrate, graphically, all aspects of the encoding model-based analysis procedure. It seems as though the authors are more or less adopting the modified implementation of forward models presented by Kok et al (2017), but the text in the methods is quite sparse. In the interest of full transparency, I recommend the authors vastly expand on this, and include additional details like specific basis function equations, etc. One potential issue w these model-based analysis methods is that the results necessarily depend on the model used for analysis, and so very clearly and transparently reporting all aspects of the model is essential. I’d recommend even adding a figure illustrating this, if possible. (minor: Eq 5 uses superscripts instead of subscripts for B₂, C₂)

We have expanded the forward encoding section in the Methods, and have included all relevant equations. We have also clearly set out how the models were tested and trained. To further clarify our approach, we have added a new figure which illustrates the modelling pictorially (Figure 9; see below).

From Page 37, Line 741

Figure 9. Schematic illustrating the forward encoding procedure used to estimate feature-selectivity for orientation in Experiment 2. **A.** A basis set of the nine channels used to model feature (orientation) selectivity. **B.** The basis set was used to find the expected response (regression coefficients) for each different RSVP item in every trial, for each EEG electrode (three electrodes are shown here for a single example participant). Three trials are shown for the corresponding gratings. **C.** Ordinary least squares regression was used to find regression weights for the orientation channels across trials for each EEG electrode (three electrodes are shown here for a single example participant). **D.** Shrinkage matrix^{11,13,61} that the weights were divided by to perform regularization, to account for correlated activity between electrodes. **E.** The regression weights were applied to predict the presented orientation. Neural activity (headmaps) from two trials, with the channel responses for those trials. Dotted lines indicate the presented orientations. **F.** Applying this procedure to each time point gives the time-course of feature-(orientation) selectivity (for one participant). Trials have been binned in 20° intervals, with the dotted lines representing the presented orientation in those trials. On the y-axis, 0 ms represents the onset of the item within the RSVP stream. Feature selectivity emerged around 75 ms after stimulus presentation. **G.** Modified Gaussian functions (equation) were used to quantify the tuning. The colours of the free parameters in the equation correspond to the relevant components of the tuning curve below.

c. There are important considerations regarding how to compare reconstructions when different estimated models are used for reconstruction (highlighted in Sprague et al, 2018; their discussion of “fixed” encoding models). This seems particularly relevant to this report, especially for the analyses presented in Figs. 6 and 7A, where parameters of fit Gaussians to encoding models are directly compared across target presentation conditions (T1, T2, distractor; correct/incorrect). Can the authors describe these model estimation procedures in more detail? For the training set, were only distractor/T1/T2 epochs chosen? Or all of them? I don’t think there’s necessarily a most ‘right’ way to do this – but there should be a bit more detail in the manuscript to help readers understand the authors’ procedures (see also code sharing recommendations, below).

The reviewer is correct – we used a fixed encoding model approach. We trained the model across all stimulus presentations (and used cross-validation), then applied these weights to test trials. We then split the results into the relevant experimental conditions and fitted the Gaussians to quantify selectivity. We have expanded the Methods section to include further details about this procedure, and have cited the recent Sprague et al. papers.

Page 41, Line 807

For the initial encoding analysis (Figure 5), to determine whether feature selectivity could be recovered for each RSVP item we used 20 encoding models (one for each item position in the stream) with 600 trials. We trained and tested each model across the entire 2250 ms of the trial to determine when feature selectivity emerged for that RSVP item. This analysis verified that each RSVP item could be encoded independently. We aligned all RSVP items across trials (N = 12,000; 600 trials by 20 items) and used a fixed encoding model for training and testing (Figures 6-8)^{64,65}. This meant we trained and tested all encoding models across all items (both targets and distractors) regardless of trial type^{12,13}.

d. Amplitude vs fidelity – the use of “amplitude” seems pretty straightforward (gain of Gaussian), but “fidelity” should be more clearly defined – it seems like the authors are using this to describe the “width” of the Gaussian (line 182)? If so, perhaps something like FWHM or width would be more appropriate? (e.g., you could, in principle, have a low amplitude but high fidelity, or low fidelity but high amplitude result, both of which feel counterintuitive with that language).

We thank the reviewer for highlighting this ambiguity. We used the term fidelity in the original manuscript to mean width, given by the standard deviation of the Gaussian. To clarify this, we have changed ‘fidelity’ to ‘width’ or ‘precision’ of neural representations, and ‘amplitude’ to ‘gain’ throughout the revised manuscript.

2. Data & code accessibility/availability statement – the authors should clearly describe, in the manuscript, how they plan to make data and analysis code available to the community. I recommend using [OSF.io](https://osf.io) and github.com to ensure persistent availability. Additionally, sharing code will improve the

transparency of the methods. I see that there's a software submission checklist included, but I can't find a reference to this in the manuscript itself. Indeed, in the reporting summary, the authors mention that code/data are available upon "reasonable request" – I encourage the authors to relax this requirement and post the code/data somewhere publicly accessible. But, in any case, this should be in the manuscript itself.

Thank you for this suggestion. We have now used OSF and GitHub to make the data and code accessible. This has been noted in a data accessibility statement in the manuscript.

Page 43, Line 851

Data availability

The EEG and behavioural data for both experiments are available at: <https://osf.io/f9g6h>. The code is available at: <https://github.com/MatthewFTang/AttentionalBlinkForwardEncoding>.

3. Statistics – In several places (e.g., lines 380-381), the authors show significant differences in one comparison (Lag 3) but not another (Lag 7), but do not report the test for interaction – the appropriate test would be a 2-way ANOVA (factors of Lag and target #). This also comes up earlier – Lines 297-307 and Fig 4, and in Fig. 6A (interaction between lag and accuracy).

We have added these ANOVAs as suggested. Note, however, that we did not include an ANOVA for Figure 6A, which reports the effect of the AB on orientation representation of the targets. This was because the existing comparison (within Lag) was between exactly matched stimulus conditions, whereas comparing Lag and accuracy would not be matched conditions. For instance, if we compared Lag 3 and Lag 7 trials, there could be an effect of the different latencies with T1. By comparing only the representations within each Lag condition for different accuracy we can be certain the only difference is from the AB. For the other ANOVAs, all of the conclusions of the original manuscript remain unchanged.

Some examples:

Page 11, Line 194

The accuracy reduction for T2 at Lags 2 and 3 was primarily linked to a reduction in gain. A 2 (Target; T1,T2) x 5 (Lag; 1,2,3,5,7) within-subjects ANOVA showed the gain parameter was affected by Target ($F(1,21)=10.00$, $p=0.005$, $\eta_p^2=0.32$) and Lag ($F(4,84)=11.66$, $p<0.0001$, $\eta_p^2=0.36$), and the interaction between these factors ($F(4,84)=7.10$, $p<0.0001$, $\eta_p^2=0.25$). Critically for our first theoretical question, the spread (width) of orientation errors was unaffected by the factors of Target ($F(1,21)=0.10$, $p=0.76$, $\eta_p^2=0.005$) or Lag ($F(4,84)=0.55$, $p=0.70$, $\eta_p^2=0.03$), or by the interaction between these factors ($F(4,84)=0.19$, $p=0.94$, $\eta_p^2=0.01$). The baseline parameter, which reflects guessing of random orientations, was also significantly affected by the factors of Target ($F(1,21)=12.72$, $p=0.002$, $\eta_p^2=0.38$) and Lag ($F(4,84)=4.82$, $p=0.002$, $\eta_p^2=0.19$), and there was a significant interaction between them ($F(4,84)=5.04$, $p=0.001$, $\eta_p^2=0.19$).

Page 16, Line 320

For the gain parameter, a 2 (Target; T1, T2) x 2 (Lag; 3,7) within-subjects ANOVA revealed significant main effects of Target ($F(1,22)=11.63$, $p=0.003$, $\eta_p^2=0.35$) and Lag ($F(1,22)=18.70$, $p<0.0001$, $\eta_p^2=0.46$), and a significant interaction ($F(1,22)=40.19$, $p<0.0001$, $\eta_p^2=0.65$). Likewise for the baseline parameter, there were significant effects of Target ($F(1,22)=8.96$, $p=0.007$, $\eta_p^2=0.30$) and Lag ($F(1,22)=12.21$, $p=0.002$, $\eta_p^2=0.36$), and a significant interaction ($F(1,22)=7.91$, $p=0.01$, $\eta_p^2=0.26$). By contrast, there were no significant main effects and no interaction for the width parameter (Target ($F(1,22)=1.19$, $p=0.29$, $\eta_p^2=0.05$; Lag ($F(1,22)=3.90$, $p=0.06$, $\eta_p^2=0.15$); interaction ($F(1,22)=0.14$, $p=0.71$, $\eta_p^2=0.006$).

Page 21, Line 403

Applying these criteria yielded the classic AB effect (Supplementary Figure 5). A 2 (Lag; 3,7) x 2 (Target; T1,T2) within-subjects ANOVA applied to these scores revealed significant main effects of Lag ($F(1,22)=199.52$, $p<0.0001$, $\eta_p^2=0.90$) and Target ($F(1,22)=8.58$, $p=0.008$, $\eta_p^2=0.28$), and a significant interaction ($F(1,22)=9.64$, $p=0.005$, $\eta_p^2=0.31$). Follow-up t-tests showed that Lag 3 accuracy was significantly lower than Lag 7 accuracy for T2 items ($t(22)=3.15$, Bonferroni $p=0.01$, $d=0.66$) but not for T1 items ($t(22)=1.97$, Bonferroni $p=0.12$, $d=0.41$).

Additionally, there doesn't seem to be much detail included about the permutation procedures used for evaluating encoding model success (Fig. 5) and differences between conditions (Fig. 6). The authors refer to a "two-tailed cluster-permutation" test (line, e.g., 363), but I'm not sure what is being permuted? Are trial labels during training shuffled for each subj, then results from such a null model compared against the real results?

The permutation procedure was a sign-based cluster permutation test implemented in the same manner as in Fieldtrip (Oostenveld, Martin, Maris, & Schoffelen, 2010). We have added full details of this method in the revised manuscript.

Page 43, Line 845

Non-parametric sign permutation tests^{67,68} were used to determine differences in the time courses of feature selectivity (Figures 5 and 6) between conditions. The sign of the data was randomly flipped ($N=20,000$), with equal probability, to create a null distribution. Cluster-based permutation testing was used to correct for multiple comparisons over the time series, with a cluster-form threshold of $p < .05$ and significance threshold of $p < .05$.

How does the analysis which looks for differences between conditions (Fig. 6) compare to that which looks at significant information (Fig. 5)?

The overall approach was very similar for the two analyses. The main difference concerned the epochs to which the modelling was applied. For Figure 5, 20 encoding models (one for each item in the RSVP stream) were separately applied across 2250 ms for 600 trials. As this analysis showed each item could be separately resolved for Figure 6, we stacked all presentations of the Gabors (600 trials x 20 items) and

applied one encoding model over 750 ms to increase power for detecting feature-selective information. All other aspects of the modelling were the same. We have further clarified these differences in the revised manuscript.

Page 41, Line 807

For the initial encoding analysis (Figure 5), to determine whether feature selectivity could be recovered for each RSVP item we used 20 encoding models (one for each item in the stream) with 600 trials. We trained and tested each model across the entire 2250 ms of the trial to determine when feature selectivity emerged for that RSVP item. This analysis verified that each RSVP item could be encoded independently. We aligned all RSVP items across trials (N = 12,000; 600 trials by 20 items) and used a fixed encoding model for training and testing (Figures 6-8)^{64,65}. This meant we trained and tested all encoding models across all items (both targets and distractors) regardless of trial type.

Finally, in Figure 3, it seems like Bonferroni correction for multiple comparisons was performed (Line 268) – what about for the similar tests shown for Fig. 2B?

These are all Bonferroni-corrected p values. We thank the reviewer for highlighting our omission in reporting this detail, which has been corrected throughout the revised manuscript.

4. Regression analysis – I'm not sure I understand this procedure, largely because it is very sparsely described in the Methods (Lines 640-646). What goes into the analysis? A circular variable of raw report angle? If so, how is the circularity of the variable accounted for? While I know most readers understand typical linear regression, in this case it can be helpful to very clearly write out what data is used to predict what, and how the authors treat the circular variables.

The regression matrix used an imaginary unit to convert the orientations to complex numbers, which were then regressed against the reported orientations (also imaginary). These details have been added to the revised manuscript, as outlined below.

Page 36, Line 716

We used a regression-based approach (see Figure 2D and 4C) to determine how targets and distractors within each RSVP stream influenced behavioural responses⁶⁰. To do this, we aligned the orientations of both distractor and target items from 4 items prior to the appearance of the target through to 9 items after the appearance of the target to construct a regression matrix of the presented orientations. The regression matrix was converted to complex numbers (to account for circularity of orientations) using Equation 2.

$$C_1 = \exp(1i C) \quad (2)$$

Where C is the regression matrix (in radians) and 1i is an imaginary unit. Standard linear regression was used to determine how the orientations of the items affected the reported orientation using Equation 3.

$$W = (C_1 C_1^T)^{-1} C_1^T R \quad (3)$$

Where R is the reported orientation (in radians). This was done separately for T1 and T2 reports, with a higher regression weight indicating the item was more influential in determining the reported orientation.

5. Role of working memory – the authors briefly mention working memory (line 511), but I'm more interested in how the maintenance of information over a relatively extended delay may have impacted behavioral performance. Especially because, in some cases (like where T2 is poorly encoded because it's missed due to the AB), the working memory load might actually be different – subjects are storing only one orientation rather than two. This isn't a criticism of the study by any means – I'm just curious if the authors could speak to how their results may also reflect changes in working memory load across conditions. I think their results show something neat, and potentially unexpected in the working memory literature – the shift from Load 1 to Load 2 does not seem to alter precision, instead only guessing (Fig. 4B).

This is an interesting point. The behavioural results do indeed suggest that only gain and guessing were affected when participants could report one or two targets (suggesting 1 or 2 targets were in visual working memory) while precision was unaffected. Furthermore, examining the neural data, the representation of T1 was unaffected by T2 accuracy. This potentially suggests that for a dual-target RSVP task, the neural representation of orientation is not modulated by working memory load. Perhaps working memory operates sequentially during temporal attention, with each item being consolidated fully before moving on to the next item. This explanation would fit with the idea that T2 fails to reach the consolidation stage during the AB. We have added a comment on this issue in the third paragraph of the revised Discussion.

Page 31, Line 597

These behavioural results may be consistent with sequential working memory consolidation of targets. We found the precision of reporting T1 was unaffected by Lag, even though often during the AB only one item is reported, whereas at longer lags two items are reported. During spatial working memory tasks, where multiple items are simultaneously presented, longer lags should have a higher memory load and lead to lower precision⁴⁷. Instead, the current results suggest that each target is consolidated into working memory before the store allows a second item to enter.

Reviewer 3

Summary

This is an interesting study that makes clever use of multivariate analyses to make a provocative point. The novelty here is the use of an encoding model to directly track changes in the fidelity of target and distractor representations during an RSVP task, which could yield new and important insights into the mechanisms responsible for the AB. Overall I found the results compelling but have some concerns about whether the conclusions are justified. The authors

should be able to address these in a revision. For clarity, I've enumerated my questions below.

1. Re: modeling the behavioral data, typically report errors are modeled with parametric models assuming that the observed data are a mixture of different trial outcomes (e.g., correctly reporting the to-be-recalled item with precision k vs. guessing; Zhang & Luck, 2008; Bays et al., 2009; Bays, 2016). The author's model captures the spirit of that approach, with the center, width, and baseline parameters capturing bias, imprecision, and guessing rates. But it's not clear what aspect of memory "amplitude" refers to, nor how it should be interpreted. Mathematically the parameter is a multiplicative scalar that controls the peak of the response relative to baseline, which implies a kind of fidelity or signal-to-noise ratio. The problem with that interpretation is that the authors are fitting a curve to a relative frequency distribution that sums to 1, and changing the baseline parameter (guessing rate) necessarily changes the amplitude (and/or width) parameter. So, what's really going on here? Is it the fidelity of the representation or the likelihood of missing T2 and guessing that's changing across lags? Both outcomes would be consistent with global workspace theory (p. 12) but would motivate very different interpretations of the data and how they map onto the fidelity of reconstructed neural representations.

We thank the reviewer for this insightful question. We chose a 4-parameter Gaussian so we could independently account for the contributions of different aspects of the AB on perception and associated orientation-selective neural responses. This function has been used in several previous papers that examined neuronal orientation selectivity (McAdams, & Maunsell, 1999, *J Neurosci*; Mazurek, Kager, & Van Hooser, 2014,, *Frontiers Neurosci*; Swindale, 1998, *Biol Cybern*; Kohn, & Movshon, 2004, *Nat Neurosci*; Gillespie, Lampl, Anderson, & Ferster, 2001, *Nat Neurosci*). As the reviewer notes, the amplitude parameter specifies the gain of the function, which is independent of width and baseline, unlike the functions used by Zhang & Luck (2008) and Bays et al. (2009, 2016). This means that if a tuning curve peaked at 1 (a.u.) and had a guessing rate of 0, the amplitude would be 1, whereas if the tuning curve peaked at 1.5 (a.u.) but the guessing rate increased to 0.5, the amplitude would still be 1. The guessing rate and the amplitude can, therefore, vary independently using this four-parameter Gaussian function. To clarify this important issue, we have re-labelled the amplitude parameter throughout the manuscript as 'gain', and have added two supplementary figures in the revised manuscript. The first figure (Supplementary Figure 1) shows how using the four-parameter Gaussian functions allows amplitude, width and baseline to vary independently. We hope this will allow readers to get an intuitive understanding of the fitted values. The second figure (Supplementary Figure 2) shows the results of a re-analysis of the data shown in Figure 2. To determine whether normalization of responses to 1 affected the results shown in Figure 2, we re-ran the analysis without normalizing the values. This yielded the same pattern of results as the original analysis. As in the original analysis, the AB decreased the gain of the orientation error distribution while correspondingly increasing baseline values, which is the guessing rate. Our results, therefore, strongly suggest that participants either had complete access to the full T2

representation (in no-AB trials) and could easily report the orientation, or had no awareness and randomly guessed the orientation in AB trials. Our neural data suggest that the T2 representation is boosted, relative to the distractors, in no-AB trials, whereas it is suppressed in AB trials. This boosting therefore appears to be associated with participants' having conscious access to the stimuli, while the suppression is associated with no awareness, which leads to guessing.

$$G(x) = A \exp\left(-\frac{(\varphi - \mu)^2}{2\sigma^2}\right) + C$$

A = Gain
 σ = Width
 C = Baseline
 μ = Centre orientation set to 90 deg

Supplementary Figure 1. Examples of the four-parameter Gaussians used to quantify behavioural orientation errors and forward encoding representations for orientation from EEG activity. The figure shows the effect of different parameter values on the shape of the resulting function. Each row has a different gain value, and each column has a different width parameter. Within each panel, the baseline value changes. The width parameter shows the precision (of either the behavioural responses or neural representations). The baseline parameter captures non-selective responses that are unrelated to the target. For the behavioural analysis, this reflects random guessing which would be distributed equally across all orientations; for the EEG analysis, it reflects overall, non-feature selective activity from the orientation encoding. For all panels, the centre orientation of the Gaussian is set to 90°. The figure highlights the independence of the parameters of the Gaussians. For instance, looking at the panels across a given row (where width varies, but gain is fixed) reveals that curves with the same baseline value have peaks at the same height. Inspecting any one panel shows that the baseline and gain parameters are independent, with the differences between the peaks of the curves being equal regardless of the baseline.

Supplementary Figure 2. Re-analysis of behavioural results in Experiment 1. In the original analysis (Figure 2), the responses were normalized to 1 for each participant to show the proportion of total responses in each orientation error bin. To confirm that this normalization did not bias the results, we re-ran the analysis but used the total number of responses (120 trials for each condition). **A.** The distribution of response errors (difference between presented and reported orientation) across participants for T1 and T2 for each Lag condition. Lines show fitted Gaussian functions. **B.** Quantified behavioural responses for the four parameters of the fitted Gaussian function for each participant. We used 2 (Target; T1,T2) x 5 (Lag; 1,2,3,5,7) within-subject ANOVAs to quantify the effect of the AB on the parameters of the fitted Gaussians. The gain parameter was affected by the factors of Target ($F(1,21)=8.97$, $p=0.007$, $\eta_p^2=0.30$) and Lag ($F(4,84)=11.78$, $p<.0001$, $\eta_p^2=0.36$), and there was a significant interaction between these factors ($F(4,84)=7.29$, $p<.0001$, $\eta_p^2=0.26$). By contrast, and consistent with the original analysis, for the width parameter there were no significant main effects of Target ($F(1,21)=0.54$, $p=0.47$, $\eta_p^2=0.02$) or Lag ($F(4,84)=1.08$, $p=0.37$, $\eta_p^2=0.05$), and no interaction ($F(4,84)=0.60$, $p=0.66$, $\eta_p^2=0.03$). The baseline parameter, which reflects guessing of random orientations, was significantly affected by Target ($F(1,21)=8.72$, $p=0.008$, $\eta_p^2=0.29$) and Lag ($F(4,84)=3.54$, $p=0.01$, $\eta_p^2=0.14$). There was also a significant interaction between these factors ($F(4,84)=3.04$, $p=0.02$, $\eta_p^2=0.13$). Taken together, the results replicate those reported in the main analysis, and confirm that the process of normalization did not bias the outcomes. Asterisks indicate Bonferroni-corrected differences at $p < .05$. Error bars indicate ± 1 standard error of mean.

2. Related: using the authors' approach the parameter estimates returned by the model will vary to some extent based on how the data are binned (e.g., Fig 2A). A likelihood-based approach that leverages the full set of report errors from each participant could yield a more stable estimate. It'd be nice to see the

outcome of such an analysis, if only to show that both approaches yield comparable results.

We thank the reviewer for this suggestion. We attempted to use maximum-likelihood estimation for the behavioural results, but this did not produce sensible values for the parameters. We instead implemented a related analysis to determine whether bin size used to characterise the orientation error distributions affected the results. To do this, we changed the bin size in steps (from 1° to 35°) and fitted four-parameter Gaussian functions to the curve associated with each bin size. We then carried out the same 5 (Lag; 1,2,3,5,7) x 2 (Target; T1,T2) within-subjects ANOVA as originally undertaken to determine whether the classic AB effect (the interaction between Lag and Target) was present for each parameter for each bin size. In line with the original results, the analysis revealed consistent effects for gain and baseline, regardless of the bin size, but no significant effects for the width or centre parameters. This suggests that bin size did not bias the original results. We have now included this additional analysis as Supplementary Figure 3 in the revised manuscript.

Supplementary Figure 3. Re-analysis of behavioural results from Experiment 1, with different bin sizes of orientation errors to generate the orientation error histogram. In the original analysis, a 15° bin size was used to group responses. **A.** Examples of Lag 2 histograms for two bin sizes (5° and 20°) for responses across participants. For the 5° bin size, orientation errors from -90° to -86° would be grouped together. By contrast, for a 20° bin, orientation errors from -90° to -71° would be grouped together. **B.** For each participant, Gaussians were fit to the resulting response function for each Lag and Target (T1 and T2). Here the fits are shown for 5° (top row) and 20° (bottom row) for the four parameters of the Gaussian. The p value is for the interaction term from the within-subjects ANOVA used in the original analysis, with factors of Lag (1,2,3,5,7) x Target (T1, T2). In the original analysis, as in the classic AB, a significant Lag x Target interaction shows that T2 accuracy is impaired at early Lags, whereas T1 accuracy is unaffected by Lag. **C.** P-values for the interaction term (Lag x Target) across a wide range of bin sizes for the four Gaussian parameters. The dotted line indicates $p = 0.05$. Note that the p values are displayed on a log axis. As in the original analysis, across this wide range of bin sizes, there was a clear AB effect on the gain and baseline parameters of the Gaussian, but no such effect on the width or centre orientation parameters.

3. The authors fit model-based reconstructions of stimulus orientation with the same function used to characterize behavioral performance and use the amplitude parameter as an index of orientation selectivity. That's fine, but it'd be helpful to know if there are differences in reconstruction bandwidth or bias across lags. If so, then the authors' selectivity index might be missing important information that could inform their conclusions. Showing that there's a difference in amplitude but not bandwidth during correct vs. incorrect trials addresses this to some extent, but it'd be helpful to see the same analysis applied to the full data set.

In the original manuscript, we included the same analysis on the width parameter as a supplementary figure. In the revised manuscript (and shown below), we re-analysed the data and included the other two parameters (centre orientation and baseline). Neither of these parameters was significantly different when comparing correct and incorrect trials. There was a tendency for the baseline parameter to increase during the AB, but this did not reach our conservative threshold for significance in the cluster analysis.

Supplementary Figure 6. Plots of three fitted parameters for the neural representations of feature-selective information over time for T1 and T2 items in Experiment 2. **A.** Time course of measured width of feature selectivity for T1 and T2 items, given by the width (standard deviation) of the fitted Gaussian. Trials were scored as correct if the participant's response was within 30° of the presented orientation. Only trials in which participants responded correctly to T1 were included in the analysis. **B.** Same as in panel **A** but for the centre orientation parameter. **C.** Same as in panel **A** but for the baseline parameter. For all panels, there were no significant differences between conditions (two-tailed cluster-permutation, alpha $p < .05$, cluster alpha $p < .05$, N permutations = 20,000). Shading indicates ± 1 standard error of mean.

4. The authors compare the relative strengths of reconstructed target and distractor representations in an attempt to adjudicate between T1-based and distractor-based accounts of the AB (p. 24-25). During correct trials, they report stronger representations of T1 and T2 relative to distractors. During incorrect trials, they report no difference in representation strength between T1 and distractors and reduced representation strength for T2 relative to distractors. That implies reduced processing of T1 and T2 during correct vs. incorrect trials, but it doesn't really tell us anything about whether the AB reflects prolonged processing of T1 or inadvertent processing of distractors. To test the T1 processing model the authors could compare T2 selectivity as a function of T1 selectivity: T2 selectivity should be reduced (and report errors higher) when T1 selectivity is high vs. low during T2 presentation. To test the

distractor model the authors could compare selectivity for distractors presented before T1 to distractors presented during T1 and T2: distractors presented during the AB should have a higher selectivity during the AB period relative to distractors presented before or after both targets.

We thank the reviewer for this thoughtful suggestion for evaluating the T1-based versus distractor models of the AB. We have capitalized on this suggestion to guide a new analysis in the revised manuscript that we believe provides the kind of strong test suggested by the reviewer. Because forward encoding results are quite noisy on a trial-by-trial basis, however, we did not feel confident to split T2 selectivity by T1 selectivity, which would require trial-by-trial fits. Instead, we expanded our existing analysis to incorporate the reviewer's idea. As the reviewer notes, selectivity should relate directly to behavioural orientation errors, so splitting the data in this way should address the reviewer's question. To evaluate the T1-based account, we compared distractor representations with target representations for correct and incorrect trials (as per our original analysis). The target-based model predicts that target representations are boosted for correct trials but suppressed for incorrect trials, as we found. To test the distractor-based model, we followed the reviewer's suggestion and compared distractor presentations before and during the AB. There was no difference in representations regardless of accuracy. This provides strong support for the target-based account of the AB.

Page 26, Line 486

For correct trials (i.e., orientation responses to T2 were within 30° of the presented orientation), the two targets resulted in significantly higher feature selectivity (gain) than the immediately adjacent distractors (-2,-1,+1 and +2 items) for both T1 and T2 representations (all $p < 0.04$). On incorrect trials, feature selectivity for T1 was not significantly greater than selectivity for the surrounding distractors ($t(22)=0.15$, $p=0.88$, $d=0.03$), even though we included only trials in which T1 was correctly reported. Most interestingly, on incorrect trials the representations of T2 items were significantly lower than that of the immediately adjacent distractors ($t(22)=2.09$, $p=0.04$, $d=0.44$), suggesting that the featural information carried by T2 was suppressed, while distractors were unaffected. To directly test the distractor model of the AB, we compared distractor representations before T1 with distractor representations during the AB (i.e., between T1 and T2). The account predicts that distractors presented during the AB should have a stronger neural representation as they are likely to be incorrectly selected as targets. Instead, we found that distractors were represented similarly before and during the AB for both correct trials ($t(22)=0.85$, $p=0.40$, $d=0.18$) and incorrect trials ($t(22)=1.83$, $p=0.08$, $d=0.38$). Taken together, these results suggest that for trials where participants accurately report target orientation, the neural representations of targets are boosted relative to those of distractors. By contrast, when the second target is missed, as occurs during the AB, there is a significant *suppression* of the target's featural information.

5. The authors' findings are broadly consistent with an "early-stage" view of the AB where impairments in T2 reports reflect impaired perceptual processing. Yet other EEG studies cited by the authors (e.g., Vogel, Luck, Shapiro) imply that stimuli can be accessed but not reported during the AB.

Can the authors square their findings with these earlier reports? Some additional discussion of this point would be helpful.

We believe our results are broadly consistent with this earlier work (as does Kimron Shapiro, the author of that account, who has seen the results of our study and has given us feedback on them) while adding to our knowledge of how the late suppression occurs. It seems likely that T2 is suppressed while T1 processing is ongoing, possibly to stop this information from being passed to higher-level areas associated with working memory consolidation. Because T2 has been suppressed, there is a smaller working memory ERP component (because there is nothing to process). We have expanded our discussion of this in the revised manuscript.

Page 30, Line 587

Suppression of the T2 representation occurred 100-150 ms after the target appeared, suggesting inhibition by ongoing processing of T1. This fits well with previous work showing that the AB is associated with a reduced late-stage response, as indicated by an ERP component associated with working memory consolidation^{8,46}. Taken together with the current results, it seems that suppressing the initial T2 representation is likely to inhibit information from being passed to later cortical stages associated with working memory consolidation. As this information does not reach higher cortical stages, it results in a reduced working memory ERP component during the AB.

6. The authors define correct trials as $\leq 30^\circ$ report error for T2 given that T1 was correctly reported (presumably “T1 correct vs. incorrect” was defined based on the same criterion? I couldn’t find this information in the manuscript). I understand that this criterion was chosen to yield an approximately equal number of correct and incorrect trials (related: it’d be helpful if the authors stated average \pm SEM correct/incorrect trials across participants somewhere in the manuscript). However, $\pm 30^\circ$ seems like a liberal threshold: if report errors are scored from -90 to 89, then the average absolute recall error for a participant who randomly guesses on each trial is $\pm 45^\circ$. Do the results change appreciably if a more stringent criterion is adopted?

This question was also asked by Reviewer 1 (see their first point). As we outline in detail above in response to this question, the representation of T2 is systematically-modulated by the behavioural accuracy, with smaller errors being associated with larger representations and large errors with small representations (Figure 7). We have expanded on this point in the revised manuscript. We have also added Supplementary Figure 4 to show the behavioural results of Experiment 2 defined by these criteria.

Page 24, Line 450

The reduction in T2 selectivity for incorrect trials at Lag 3 was not driven by an arbitrary split of trials into correct and incorrect categories. To verify this, we sorted the evoked T2 forward encoding results by the amount of orientation error (in 15° error bins to allow sufficient signal-to-noise ratios for fitting). There was significantly greater feature selectivity when the orientation error was small, and this selectivity

decreased gradually with larger errors (one-way within-subjects ANOVA, $F(1,22)=2.76$, $p=0.02$, $\eta_p^2=0.11$; Figure 7). These results indicate that the AB is associated with a reduction in gain, but not width, of feature-selective information for the second target item (T2), and that this effect occurs soon after the target appears within the RSVP stream.

Figure 7. Gain of feature-selective information for T2 items presented at Lag 3 in Experiment 2, plotted as a function of reporting error. Forward encoding results were averaged across early time points (100-150ms), and were binned by the absolute difference between the presented and reported orientations (in 15° increments). Each bin is displayed as the starting value (e.g., 0° incorporates errors from 0° to 15°). Gaussians were fitted to quantify selectivity with the gain parameter shown here. Feature selectivity was highest when participants reported the orientation to within 30° of the presented orientation, and declined significantly with larger reporting errors. Error bars indicate ± 1 standard error of the mean.

Reviewers' Comments:

Reviewer #1:

Remarks to the Author:

Point 1.

This additional analysis did confirm that the reduction in feature selectivity in T2 was associated with errors in behavior. That said, this linear pattern seems to complicate the interpretation as well, If I understand correctly – that is, as feature selectivity gain gradually decreased, the reporting errors increased linearly (at least for a range between 0-45 degree). This seems more consistent with a graded account of AB.

Point 2.

Supplementary Figure 4 present a clear way to demonstrate the switching errors: a strong correspondence between presented T2 and reported T1. There is a minor typo in caption to "Supplementary Figure 4": "The rightmost column shows T2 switching, where presented-T2 orientation is plotted against reported-T2 orientation", should be "reported-T1 orientation".

Point 3.

Supplementary Figure 1 presents the independence of the 4 parameters used in the current study, but does not demonstrate why it is superior as compared to the other models. Asplund et al. model could not independently estimate the gain and precision of the response?

Point 4.

The explanation is okay to me. Perhaps the authors could perform a similar analysis for Lag7, as they did for Lag3 in Figure 6, to show that the behavioral errors at Lag7 (i.e., divided trials into correct and incorrect) was not due to the loss of the feature selectivity gain (i.e., undistinguished EEG profile.)

Point 5.

Okay.

Point 6.

The explanation is not clear to me. The authors attributed the effect on T2 representations between 100-150ms, which they argued as the neural correlates of the AB, to the suppression of its early sensory processing, but cited previous studies which argued that the AB is due to the late-stage processing associated with working memory consolidation. Then, why the early sensory processing on T2 has to be suppressed to stop inference from T1 at late-stage processing (e.g., working memory consolidation). This explanation makes the neural locus of the AB unclear.

Point 7.

Okay to me.

Point 8.

I don't think the authors directly address this point. It is still not clear why they included Lag 3, instead of Lag 2, in Exp.2.

Minor comments:

Point 1.

When the analysis was restricted on trials in which T1 was correct (i.e., within 30 degree), the behavioral results (Supplementary Figure 5) were biased, as compared to Figure 4A. In Supplementary Figure 5, for Lag7, was accuracy on T2 significantly lower than T1? It seems so, as indicated by a significant main effect of Target. The authors only reported follow-up comparisons

between lag2 and lag7 in each Target condition, but not the other way.

Point 2.

Not addressed. If there was no restriction on the difference in orientations of T1 and T2, I cannot see why repetition blindness, which has a different neural locus as compared to that for the AB, could be ruled out in the current study.

Point 3.

Okay to me.

Reviewer #2:

Remarks to the Author:

I appreciate the authors' careful consideration of my comments in their revision. I am very satisfied with the resubmitted manuscript, and have no further concerns. I think this will be an important report of substantial interest to the readership of Nature Communications.

I found a few very small corrections the authors may wish to make:

1. The authors may wish to double-check the units of Fig. 6B – the values seem quite different from those in Fig. 5. If these are indeed correct, a brief comment justifying their difference in the figure caption might be helpful to avoid reader confusion
2. Supplementary Fig. 4, caption, Line 941 – I believe the description of the rightmost column may include a typo. I think the authors mean to write "against reported T1 orientation" (based on the axes labels of the figure).

Reviewer #3:

Remarks to the Author:

The authors have done a commendable job of replying to reviewer comments, and have successfully addressed my earlier comments. I have no additional concerns, and congratulate the authors on a nice paper.

Response to reviewers

We are pleased that Reviewers 2 and 3 were satisfied with the changes implemented in our initial revision. We have now addressed the remaining issues raised by Reviewer 1, and trust the manuscript is now suitable for publication in *Nature Communications*.

In the text that follows, reviewer comments are shown **in bold** and our responses to them are given immediately below in normal font. For convenience, we have reproduced major changes in the manuscript text, verbatim, **in red font**.

Reviewer #1

Point 1.

This additional analysis did confirm that the reduction in feature selectivity in T2 was associated with errors in behavior. That said, this linear pattern seems to complicate the interpretation as well, If I understand correctly – that is, as feature selectivity gain gradually decreased, the reporting errors increased linearly (at least for a range between 0-45 degree). This seems more consistent with a graded account of AB.

Examination of Figure 7 reveals that the largest orientation gain was present when behavioural errors were less than 29 degrees (recall that the 15-deg condition included orientations from 15-29 degrees), and that gain decreased with larger errors. Rather than supporting a graded account of the AB, this outcome simply reflects the nature of the task, which required that participants report orientation on a continuous scale (rather than on a categorical scale, as is the case in many conventional AB studies, e.g., Raymond, et al., 1992; Vogel, et al., 1998). As guessing rates increase, responses become more variable and this inevitably leads to larger orientation errors. The results throughout the study, in conjunction with previous findings (Asplund et al., 2014), clearly support a discrete, not graded, model of the AB. We have noted this in the revision (Page 24, Line 462).

Note that this does not suggest a graded model of the AB, but is consistent with response variability increasing with guessing. This finding is consistent with the behavioural results, which suggest a discrete model of the AB.

Point 2.

Supplementary Figure 4 present a clear way to demonstrate the switching errors: a strong correspondence between presented T2 and reported T1. There is a minor typo in caption to “Supplementary Figure 4”: “The rightmost column shows T2 switching, where presented-T2 orientation is plotted against reported-T2 orientation”, should be “reported-T1 orientation”.

Thank you for noting this typo. We have corrected it in the revision (Page 48, Line 963).

Point 3.

Supplementary Figure 1 presents the independence of the 4 parameters used in the current study, but does not demonstrate why it is superior as compared to the other models. Asplund et al. model could not independently estimate the gain and precision of the response?

The function used by Asplund et al. (based on the original Bays et al. studies) conflates width and gain (the concentration parameter), which was acceptable for their purposes (and for Bays et al., 2009), where the aim was to model working memory consolidation; but the function is not appropriate for the current study, which aimed to determine whether the AB affects the gain or width on selectivity. We have included a figure below to help demonstrate the effect of different parameters on the function used by Asplund et al. The guessing rate is varied within each panel, and the concentration parameter is varied between the panels. This figure demonstrates that with the Asplund et al. approach, as the concentration parameter increases, the gain also increases and the width decreases. The estimates of gain and width are, therefore, conflated using this function. Furthermore, and perhaps most importantly, using the same four-parameter Gaussian function for behavioural and neural data, which would not be possible with the Asplund function, allowed us to directly compare the different measures. We have included a discussion of these issues in the revision (Page 37, Line 731).

We used a Gaussian with a constant offset to characterise behavioural performance, as it captures the distribution of errors well (median $R^2 = 0.76$, $SE = 0.04$ in Experiment 1). This model allows the gain, width, bias and guessing rates to vary independently (Supplementary Figure 1), unlike the function used in a previous study using a continuous report measure for the AB²⁷. Most importantly, the function we implemented can also be used to characterise the forward encoding results, thus allowing a direct comparison of the AB based upon behavioural and neural measures.

Point 4.

The explanation is okay to me. Perhaps the authors could perform a similar analysis for Lag7, as they did for Lag3 in Figure 6, to show that the behavioral errors at Lag7 (i.e., divided trials into correct and incorrect) was not due to the loss of the feature selectivity gain (i.e., undistinguished EEG profile.)

We are pleased that our response to this point was satisfactory. Note, however, that the current version of Figure 6 already shows the effect of behaviour on the neural representation of Lag 7 items. In Figure 6A, the bottom row shows feature selectivity based on reporting accuracy for T1 and T2 at Lag 7 (left and right panels, respectively). There was no significant effect of behaviour (correct vs. incorrect responses) on gain representations for either T1 or T2 items across any time points. In the revision, we have further quantified the results in Figure 6B, showing that neither the gain nor the width of the Lag 7 representation was affected by behavioural responses (correct vs. incorrect) when we consider the forward encoding results averaged over 100-150 ms (see Page 24, Line 452).

For Lag 7 items, neither the gain ($t(22)=0.12$, $p=0.90$, $d=0.03$; Figure 6B lower panel) nor the width ($t(22)=0.04$, $p=0.96$, $d=0.01$) of the neural representations of T2 items were affected by behavioural performance (correct vs. incorrect trials).

Point 5.

Okay.

Point 6.

The explanation is not clear to me. The authors attributed the effect on T2 representations between 100-150ms, which they argued as the neural correlates of the AB, to the suppression of its early sensory processing, but cited previous studies which argued that the AB is due to the late-stage processing associated with working memory consolidation. Then, why the early sensory processing on T2 has to be suppressed to stop inference from T1 at late-stage processing (e.g., working memory consolidation). This explanation makes the neural locus of the AB unclear.

We would first like to clarify our argument. In R1's initial review, a question was raised as to why there was suppression of early (100-150ms) feature selectivity during the AB in Experiment 2. Recall that we employed forward encoding modelling to extract the neural representations of orientation-information of stimulus features. During the AB, we observed significant suppression of this information shortly after T2 presentation. As the forward encoding analyses reveal neural activity exclusively associated with relevant stimulus features (in this case, orientation selectivity), we can conclude that it is this information that is suppressed during the AB. We argued that the early sensory component is suppressed to prevent the T2 representation from interfering with ongoing T1 processing. We noted that a previous study by Vogel, et al. (1998) found reduced late-stage working memory (WM) consolidation during the AB. Our results are broadly consistent with this finding because if the target has already been suppressed at an early stage, then there will necessarily be less information to consolidate into WM, and thus a corresponding reduction in the

associated ERP component. Having made this point, any direct comparison between our results and those of Vogel et al. (1998) must be made with caution, given the differences between the tasks, stimuli and analytic approaches across the two studies. We have clarified our explanation of this issue in the revision (Page 31, Line 600).

Taken together with the current results, it appears that the AB is associated with an early suppression of sensory information associated with the T2 stimulus. The diminished strength of sensory information associated with the T2 item in turn is expected to exert less influence on later stages in the information processing hierarchy, such as working memory.

Point 7.

Okay to me.

Point 8.

I don't think the authors directly address this point. It is still not clear why they included Lag 3, instead of Lag 2, in Exp.2.

We understand the reviewer's point but can only reiterate that the AB effects observed at Lags 2 and 3 in Experiment 1 were both substantial and only marginally different from one another. We certainly could have chosen Lag 2 instead of Lag 3 for inclusion in Experiment 2. As the AB was slightly larger at Lag 3, it is likely the already significant and robust results from Experiment 2 would have been larger still. The use of Lag 3 (instead of Lag 2) to probe the AB is common in the literature (see, for example, Ghorashi, Spalek, & Lollo, 2009; Harris, McMahon, & Woldorff, 2013; Janson & Kranczioch, 2014a; 2014b; Lagroix, Spalek, Wyble, Jannati, & Di Lollo, 2012; MacLean, Arnell, & Cote, 2012; Ronconi, Pincham, Szűcs, & Facoetti, 2015; T. A. W. Visser et al., 2004; 2014; Wierda, van Rijn, Taatgen, & Martens, 2010; Willems, Herdizin, & Martens, 2015; Zauner et al., 2012), which facilitates comparison of our results with those reported previously, and shows our choice of Lag 3 was not atypical. We have included a statement in the revision pointing to the use of Lag 3 in the AB literature (Page 34, Line 671).

In Experiment 2 we selected Lag 3 as the test condition for the AB because it yielded a significant reduction in T2 response accuracy compared with T1 in Experiment 1, and because it has been widely used in previous studies of the AB^{37,38,45,55-63}.

Minor comments:

Point 1.

When the analysis was restricted on trials in which T1 was correct (i.e., within 30 degree), the behavioral results (Supplementary Figure 5) were biased, as compared to Figure 4A. In Supplementary Figure 5, for Lag7, was accuracy on T2 significantly lower than T1? It seems so, as indicated by a significant main effect of Target. The authors only reported follow-up comparisons between lag2 and lag7 in each Target condition, but not the other way.

We thank the reviewer for drawing our attention to this issue. There was a coding error in our revised Supplementary Figure 5, which we have now corrected. As can be seen, the behavioural results displayed in Supplementary Figure 5 are in fact consistent with those shown in Figure 4. We have included the corrected figure below and revised the text in the revision (Page 21, Line 402).

Applying these criteria yielded the classic AB effect (Supplementary Figure 5). A 2 (Lag; 3,7) x 2 (Target; T1,T2) within-subjects ANOVA applied to these scores revealed significant main effects of Lag ($F(1,22)=19.05$, $p<0.0001$, $\eta_p^2=0.46$) and Target ($F(1,22)=18.00$, $p<0.0001$, $\eta_p^2=0.45$), and a significant interaction ($F(1,22)=31.91$, $p<0.0001$, $\eta_p^2=0.59$). Follow-up t-tests showed that Lag 3 accuracy was significantly lower than Lag 7 accuracy for T2 items ($t(22)=5.20$, Bonferroni $p=0.0001$, $d=0.44$) but not for T1 items ($t(22)=2.11$, Bonferroni $p=0.09$, $d=0.44$). In addition, T2 accuracy was significantly lower than T1 accuracy at Lag 3 ($t(22)=5.94$, Bonferroni $p<0.0001$, $d=1.08$), but there was no such difference at Lag 7 ($t(22)=1.20$, Bonferroni $p=0.48$, $d=0.25$).

Point 2.

Not addressed. If there was no restriction on the difference in orientations of T1 and T2, I cannot see why repetition blindness, which has a different neural locus as compared to that for the AB, could be ruled out in the current study.

The orientations of the target Gabors were uncorrelated over trials as they were drawn (without replacement) from a random distribution. This means the same orientation was never repeated in a trial. As repetition blindness only occurs when targets repeat in identity (see Kanwisher (1987), *Cognition*, 27, 117-143) it means, by definition, that repetition blindness could not occur. Even if repetition blindness were to occur for *similar* identities (in this case, for similar orientations), the uncorrelated nature of the task means Gabors were equally likely to be similar or dissimilar, which would cause any repetition effects to cancel out over trials. We have now noted this point in the revision (Page 33, Line 652).

On each trial, the orientations of the twenty Gabors in the stream were drawn pseudo-randomly, without replacement, from integer values ranging from 0-179°.

Both targets and distractors were drawn from the same random distribution, meaning there was no restriction on the relationship between targets (except they could not be identical). Note the uncorrelated nature of the targets means the design controls for possible repetition blindness effects⁵⁴, since the targets were equally likely to be similar in orientation as they were to be maximally dissimilar (i.e., orthogonal), and thus any potential orientation-specific effects would cancel out across trials.

Point 3.
Okay to me.

Reviewer #2

I found a few very small corrections the authors may wish to make:

1. The authors may wish to double-check the units of Fig. 6B – the values seem quite different from those in Fig. 5. If these are indeed correct, a brief comment justifying their difference in the figure caption might be helpful to avoid reader confusion.

The values in Figures 5 and 6 are correct. The difference in magnitude comes from the increased number of training trials (12,000 vs 600). We have noted this in the revision (Page 23, Line 436).

Note that the difference in magnitude for the encoding results shown in Figure 5 is due to the increased number of training trials used in this analysis (12,000 vs 600).

2. Supplementary Fig. 4, caption, Line 941 – I believe the description of the rightmost column may include a typo. I think the authors mean to write “against reported T1 orientation” (based on the axes labels of the figure).

Thank you for picking up this typo. We have corrected this in the revision.

Reviewers' Comments:

Reviewer #1:

Remarks to the Author:

Point 1.

I am okay with authors' explanation that the pattern in Figure 7 reflects the nature of the task. However, it seems that authors interpret the gain (y-axis in Figure 7) as guessing rates, but in the main text (e.g. p.11), authors regarded the baseline parameter as guessing. The authors should clarify this apparent incongruity in the paper.

Point 8.

I do not doubt the validation of using Lag 3 to demonstrate the AB in Exp. 2, and would agree that in many other studies Lag3 has been used to show the AB. But I am disappointed by the authors' explanation that "As the AB was slightly larger at Lag 3, it is likely the already significant and robust results from Experiment 2 would have been larger still.", since they could simply have analyzed lag 2. If they are right, it would have only substantiated their claims had they also analyzed Lag 2. I don't see this as a dealbreaker, however.

Response to reviewer comments:

1. I am okay with authors' explanation that the pattern in Figure 7 reflects the nature of the task. However, it seems that authors interpret the gain (y-axis in Figure 7) as guessing rates, but in the main text (e.g. p.11), authors regarded the baseline parameter as guessing. The authors should clarify this apparent incongruity in the paper.

We are pleased to have an opportunity to clarify this issue. Recall that in our analyses of the behavioural data in Experiments 1 and 2, we fit a four-parameter Gaussian to the distribution of participants' responses and to the forward encoding results from Experiment 2. One of the four parameters is "Gain" (corresponding to the amplitude of the fitted Gaussian) and another is the "Baseline" (a constant that accounts for non-orientation selective responses, i.e., guesses). Figure 7 shows that in Experiment 2 the gain parameter for feature-selectivity from forward encoding at Lag 3 is largest for the most accurate orientation judgements, and gets smaller as orientation judgements get less accurate. For the behavioural modelling we found guessing, reflected in the baseline parameter, was also affected. Specifically, in Experiment 1 the guessing rate was significantly elevated for T2 at Lags 2 and 3; and in Experiment 2 the guessing rate was significantly elevated for T2 at Lag 3. Thus, we found significant modulations of both the amplitude of feature selectivity of T2 items **and** the guessing rates associated with these items across the two experiments.

We have made a small but important modification to the text in the Results section of Experiment 2– as highlighted below – to further clarify the idea that participants' response variability is captured by significant changes in **both** the gain and baseline parameters (corresponding to feature-selective amplitude and random guessing, respectively).

Note that this finding is inconsistent with a graded model of the AB, and instead supports the idea that response variability during the AB is associated with both a decrease in feature-selective gain and an increase in the rate of guessing.

2. I do not doubt the validation of using Lag 3 to demonstrate the AB in Exp. 2, and would agree that in many other studies Lag3 has been used to show the AB. But I am disappointed by the authors' explanation that "As the AB was slightly larger at Lag 3, it is likely the already significant and robust results from Experiment 2 would have been larger still.", since they could simply have analyzed lag 2. If they are right, it would have only substantiated their claims had they also analyzed Lag 2. I don't see this as a dealbreaker, however.

We apologise that our previous response to this query was not sufficiently clear. The confusion seems to arise from R1's assumption that in Experiment 2 we collected

data on performance at Lag 2 but simply chose not to analyse these data (from R1's review, above: "...it would only have substantiated their claims had they also analysed Lag 2"). To be clear: **We did not collect data for Lag 2 trials in Experiment 2.** Thus, there simply are no relevant data to analyse. Including fewer lags in Experiment 2 increased the number of trials per lag to 300 (instead of 120 in Experiment 1), providing a higher signal-to-noise for the regression-based analysis. We have made this point twice in the manuscript, as follows:

The method [for Experiment 2] was identical in all respects, except that we now included targets only at Lags 3 and 7 (i.e., a single target inside and outside the AB, respectively) to increase the within-subject power for the EEG analyses.

There were 600 trials in each of the two experiments, with an equal distribution of trials across the lag conditions (120 in Experiment 1, 300 in Experiment 2), with fewer lags included in Experiment 2 to increase signal-to-noise for the regression-based EEG analysis. In Experiment 2, we selected Lag 3 as the test condition for the AB because it yielded a significant reduction in T2 response accuracy compared with T1 in Experiment 1, and because it has been widely used in previous studies of the AB^{24,39,40,54-57}.